# A Propagation Property of the Pareto Order with Applications to Multiobjective Optimization

## Abstract

We establish a propagation property of strict Pareto dominance $\prec$ on $\mathbb{R}^k$ and show its algorithmic consequence for Pareto archive maintenance in multiobjective optimization. Specifically, if $u, v \in \mathbb{R}^k$ are incomparable under $\prec$, then no $q \in \mathbb{R}^k$ can satisfy $u \prec q \prec v$ or $v \prec q \prec u$. We give algebraic and geometric proofs, the latter via containment of strict down-sets (lower orthants). As a corollary, we obtain a *post-witness $u \prec q$ pruning rule* for insertion into a mutually nondominated Pareto archive $S$: once a witness $w \in S$ with $q \prec w$ is found, no remaining $u \in S \setminus \{w\}$ can dominate $q$. Accordingly, all subsequent "$u \prec q$?" checks may be omitted, while "$q \prec u$?" checks are retained to remove points dominated by $q$. We provide pseudocode for the resulting single-pass archive-insertion routine and prove its correctness. To contextualize the rule's potential effect, we derive, under a standard random-input model in which points are drawn independently from a continuous distribution on $\mathbb{R}^k$, a closed-form baseline for Pareto comparability in dimension $k$, and we interpret it heuristically as a proxy for the fraction of post-witness "$u \prec q$?" checks that can be skipped once a first witness $w$ with $q \prec w$ has been found. We complement this comparison-count analysis with a small synthetic experiment showing that the observed savings track two mechanism-level quantities: witness frequency and first-witness position. Finally, we discuss extensions beyond exact strict Pareto comparisons, including weak and $\varepsilon$-dominance, certified dominance under uncertainty, and repeated-confirmation noisy-comparator models that provide local probabilistic error control under explicit assumptions.

## 1 Introduction

This work frames optimization through the lens of order theory. A minimization problem[1] is thus defined by a feasible set $\mathcal{F} \subseteq \mathcal{X}$, an objective space $Y$ equipped with a (weak) preference order $\preceq$, and an objective map $F : \mathcal{X} \to Y$ that evaluates each decision. A feasible decision $x^\star \in \mathcal{F}$ is a *minimizer* precisely when its outcome $F(x^\star)$ is a *minimal element* of the attainable outcome set $F(\mathcal{F}) \subseteq Y$ with respect to the preference order $\preceq$. This order on outcomes pulls back to the decision space, inducing a corresponding order $\preceq$ on $\mathcal{F}$:

$$x \preceq y \iff F(x) \preceq F(y) \qquad (x, y \in \mathcal{F}).$$

Equivalently, then, a decision $x^\star$ is a minimizer if and only if it is a minimal element of $\mathcal{F}$ under this induced order. Thus, optimality is fundamentally *minimality on outcomes*, transferred to the decision space by the objective map $F$.

This perspective immediately clarifies why the solution set of minimizers to $F$ may be a *singleton* or a larger set in terms of cardinality. The distinction hinges on whether the preference order $\preceq$ on $Y$ is total or partial.

When $\preceq$ is a total order on $Y$, as in standard single-objective optimization where the objective space $Y = \mathbb{R}$ is equipped with ($\preceq = \leq$), all outcomes are comparable. This property ensures that if the infimum

---

[1]We restrict attention to minimization problems without loss of generality, since maximization is equivalent under the reversed (dual) order.

of the attainable set $F(\mathcal{F})$ is reached, the minimal value is unique. While the minimal value is unique, the corresponding *argmin set* of decisions may nevertheless contain multiple points:

$$\text{argmin}_{x \in \mathcal{F}} F(x) := \{ x \in \mathcal{F} : F(x) = \min F(\mathcal{F}) \}$$

These are *ties*: distinct decisions that achieve the same optimal scalar value. In contrast, when $\preceq$ is only a partial order on $Y$, the solution concept becomes inherently set-valued. This is a characteristic of multi-objective optimization, where one seeks to minimize a set of $k$ conflicting scalar-valued objective functions $(f_1, f_2, \ldots, f_k)$ simultaneously. The objective map is then a vector of these functions $F(x) = (f_1(x), f_2(x), \ldots, f_k(x))$, and the objective space $Y = \mathbb{R}^k$ is equipped with the *product (coordinatewise) order*—also called the Pareto order [2].

This order permits *incomparable* outcomes: vectors $u, v \in Y$ that neither $u \preceq v$ nor $v \preceq u$ holds. Such incomparability signifies a fundamental trade-off, where one outcome is superior in some objectives but inferior in others. Consequently, the set $F(\mathcal{F})$ typically possesses not a single least element, but a family of minimal elements. This family is called the Pareto front, and its preimage in decision space $\mathcal{F}$ is the Pareto set, whose elements are the efficient decisions[3].

In practical multiobjective optimization, a standard computational goal is to construct a finite, reasonably faithful approximation of the Pareto front, that is, a finite, well-distributed set of outcomes intended to be representative of the typically infinite set of minimal elements of $F(\mathcal{F}) \subseteq \mathbb{R}^k$.

Conceptually, the realization of this goal involves two logically distinct tasks. The first is to generate outcome vectors by selecting feasible decisions $x_1, \ldots, x_N \in \mathcal{F}$ (for some finite $N$) and evaluating $F(x_i) \in \mathbb{R}^k$, thereby producing the finite observed outcome set $T := \{F(x_1), \ldots, F(x_N)\} \subseteq F(\mathcal{F})$. The second is to perform *order-processing* on $T$. In particular, one may maintain a set $S \subseteq T$ intended to represent the minimal elements within $T$ (in the simplest case, $S$ is exactly that set). Two recurrent routines arise within this order-processing step:

(i) *archive maintenance*, which updates $S$ upon observing an additional outcome vector $z \in \mathbb{R}^k$ by first augmenting the observed set $T \leftarrow T \cup \{z\}$ and then updating $S$ so that it remains the set of minimal elements of $T$ (removing elements that cease to be minimal and retaining $z$ only if it is minimal in the updated set); and (ii) *front-by-front decomposition* (nondominated sorting), which decomposes $T$ into a hierarchy of Pareto layers through iterated minimal-element extraction: it extracts the minimal elements of $T$, removes them, and repeats this operation on the remainder until $T$ is exhausted Deb et al. (2002); Jensen (2003); Zhang et al. (2014).

The primitive operation in both routines is the pairwise comparison of two outcome vectors in $\mathbb{R}^k$ under the Pareto order. For a finite set of $|T|$ outcome vectors, naive implementations may require on the order of $|T|^2$ such comparisons, and each comparison may inspect up to $k$ coordinates (often fewer due to early termination). Consequently, the order-processing workload can become substantial when the number of processed outcomes $|T|$ and/or $k$ grows Deb et al. (2002); Jensen (2003). For this reason, counts of pairwise order comparisons are widely used as an implementation-independent proxy for the workload of sorting/archiving routines, and are commonly reported alongside wall-clock time when comparing nondominated-sorting variants Zhang et al. (2014); Long et al. (2021).

Thus, it remains valuable to isolate and exploit structural consequences of the Pareto order that allow many pairwise order relations to be inferred from a small number of explicit comparisons. The present paper isolates one such consequence and shows that it yields a correctness-preserving pruning rule for the order-processing routines above.

## 1.1 Contributions

The contributions of the paper are as follows:

---

[2]In what follows, all references to minimal elements and minimality are with respect to the Pareto order unless stated otherwise

[3]See Section 2 for formal definitions and notation.

1. **A propagation property under strict Pareto dominance.** We formalize and prove a propagation property of strict Pareto dominance: if $u \parallel v$, then there is no $q$ such that $u \prec q \prec v$ or $v \prec q \prec u$. This is stated in Theorem 1 and equivalently in the "no two-step bridge" statement in Corollary 1. We also present algebraic, transitivity-based, and geometric proofs that make explicit the order-theoretic mechanism used later in the paper.

2. **Algorithmic consequence for single-pass archive insertion.** Specializing the propagation property to a mutually nondominated archive yields a one-sided pruning rule for single linear-scan archive insertion: once a witness $w$ with $q \prec w$ has been found, no later archive element can satisfy $u \prec q$. This is stated in Corollary 2 and, for the archive setting, in Corollary 3. We give a drop-in insertion routine (Algorithm 1) and prove its correctness (Theorem 2). The rule preserves the worst-case $O(m)$ complexity of linear-scan insertion while reducing dominance-predicate calls whenever a witness appears before the end of the scan.

3. **Quantitative proxy-model analysis under random inputs.** Under an independent and identically distributed (i.i.d.) general position continuous sampling model, we derive a closed-form comparability/incomparability calculation (Proposition 1) that serves as a stylized proxy baseline for the prevalence of local post-witness pruning opportunities in unstructured random point sets. This calculation is not a direct model of curated archive dynamics, but rather a reference point for interpreting how incomparability scales with the ambient dimension $k$.

4. **Empirical illustration of dominance-predicate savings.** We include a small synthetic experiment comparing archive insertion with and without the post-witness pruning rule. Its purpose is explanatory rather than confirmatory: it makes the resulting reduction in dominance-predicate calls directly observable under controlled insertion streams, and relates the observed savings to witness frequency and first-witness position under different scan orders.

5. **Extensions beyond exact strict Pareto comparisons.** We discuss how the same order-theoretic viewpoint extends beyond exact strict Pareto order. First, we note that the same exclusion logic applies to weak and $\varepsilon$-dominance whenever the operative pruning relation is transitive and incomparability is defined with respect to that same relation (Remark 4). Second, for noisy objective evaluations, we consider two distinct extensions: a certified-dominance framework based on uncertainty bounds, which yields conditional exactness on the relevant coverage event, and a repeated-confirmation noisy-comparator model, which yields local probabilistic error control under explicit assumptions. We discuss safeguards against catastrophic archive elimination and mechanisms for correcting noise-induced deletions.

## 2 Preliminaries: orders, dominance, and down-sets

This section fixes notation and recalls the order-theoretic notions used throughout: (pre)orders, dominance relations on outcomes and decisions, and down-sets (order ideals). We keep the presentation self-contained and aligned with the optimization viewpoint.

For basic order theory we follow Davey & Priestley (2002); Grätzer (2011); for multiobjective optimization and Pareto efficiency, Miettinen (1999); Ehrgott (2005); algorithmic context draws on Deb (2001); Zhang et al. (2014).

### 2.1 Orders and preorders

A *binary relation* $\preceq$ on a set $Y$ is a subset of $Y \times Y$. We write $u \preceq v$ for $(u, v) \in \preceq$ and define its *strict part* by

$$u \prec v \iff \big(u \preceq v \,\wedge\, \neg(v \preceq u)\big).$$

A relation $\preceq$ is a

- *preorder* if it is reflexive and transitive;

- *partial order* (poset) if it is a preorder and antisymmetric ($u \preceq v$ and $v \preceq u \Rightarrow u = v$);

- *total (linear) order* if it is a partial order and *complete* ($u \preceq v$ or $v \preceq u$ for all $u, v \in Y$).

Given a preorder $\preceq$, the *indifference* relation $u \sim v \iff (u \preceq v \ \land \ v \preceq u)$ is an equivalence relation; the quotient $Y/\sim$ with the induced relation is a poset.

For $S \subseteq Y$:

- $m \in S$ is *minimal* if there is no $s \in S$ with $s \prec m$;

- $\ell \in S$ is *least* if $\ell \preceq s$ for all $s \in S$.

Every least element is minimal; the converse need not hold in partial orders.[4] We write $Min(S)$ for the set of all minimal elements of $S$.

## 2.2 Product orders on $\mathbb{R}^k$

The canonical order on $\mathbb{R}^k$ used in multiobjective optimization is the *product (coordinatewise) order*. This is a partial order $\preceq$ defined by componentwise inequality:

$$\forall\, u, v \in \mathbb{R}^k: \quad u \preceq v \iff \forall\, i \in \{1, \ldots, k\},\ u_i \leq v_i, \tag{1a}$$

$$u \prec v \iff (u \preceq v \ \land \ u \neq v)$$

$$\iff \left(\forall\, i,\ u_i \leq v_i\right) \ \land \ \left(\exists\, j,\ u_j < v_j\right). \tag{1b}$$

Here equation 1b is the strict counterpart of the weak-order equation 1a.

## 2.3 Dominance on outcomes and on decisions

Let $(Y, \preceq)$ be the objective space with $Y = \mathbb{R}^k$ endowed with the canonical coordinatewise *partial* order $\preceq$ in equation 1, and let $F: \mathcal{X} \to Y$ be the objective map. This order encodes the outcome–level notion of "no worse than" (weak dominance). Pulling it back to the decision space defines a *preorder* on decisions:

$$x \preceq y \iff F(x) \preceq F(y), \qquad x \prec y \iff (x \preceq y \ \land \ \neg(y \preceq x)). \tag{2}$$

Commonly, this coordinatewise product order is called Pareto order. Under this order, for any two vectors $u, v \in Y$ exactly one of the following four mutually exclusive relations holds:

- **Strict dominance**: $u \prec v$ (read as "$u$ dominates $v$"). Formally, $u \prec v \iff (u \preceq v \ \land \ u \neq v)$. This means $u$ is better than or equal to $v$ in all objectives $\{f_i\}_{i=1}^k$ and strictly better in at least one.

- **Inverse strict dominance:** $v \prec u$ (read as "$u$ is dominated by $v$").

- **Equality:** $u = v$. Note that for this partial order, indifference ($u \preceq v \land v \preceq u$) is equivalent to equality.

- **Strict incomparability:** $u \parallel v$. This holds if neither $u \prec v$ nor $v \prec u$ is true. Incomparability represents a trade-off: one vector is strictly better in at least one component, while the other is strictly better in at least one different component.

---

[4]In a total order, any nonempty $S$ has *at most* one minimal element; if it exists, it is automatically the least element. In partial orders, $S$ can have many minimal elements.

**Pareto efficiency and fronts.** A feasible point $x^\star$ is (Pareto) *efficient* or Pareto optimal/minimal if there is no $x \in \mathcal{F}$ with $x \prec x^\star$, equivalently $F(x^\star) \in Min\big(F(\mathcal{F})\big)$ under $\prec$. The *Pareto set* is

$$\mathcal{P} \;=\; \Big\{ x \in \mathcal{F} : \; \nexists\, y \in \mathcal{F} \text{ with } y \prec x \Big\},$$

and its image $F(\mathcal{P})$ is the *nondominated Pareto front.*

**Standing convention.** From now on, unless stated otherwise, "Pareto dominance" means the *strict* relation $u \prec v$. "Incomparable" means *strict incomparability*, denoted $u \parallel v$, i.e., $u \nprec v$ and $v \nprec u$. When the non-strict order is intended, we will write $u \preceq v$, and $u \parallel_{\preceq} v$ explicitly.

### 2.4 Down-sets (order ideals) and strict down-sets

Let $(Y, \preceq)$ be a poset and write its strict part by $x \prec y \iff (x \preceq y$ and $x \neq y)$. For $u \in Y$:

$$\downarrow u \;:=\; \{\, v \in Y : v \preceq u \,\} \qquad \text{(weak/principal down-set)},$$

$$\downarrow_{\prec} u \;:=\; \{\, v \in Y : v \prec u \,\} \qquad \text{(strict down-set)}.$$

Clearly,

$$\downarrow_{\prec} u \;=\; \downarrow u \setminus \{u\}.$$

**Closure under moving downward.** If $q \in\, \downarrow u$ and $w \preceq q$, then $w \in\, \downarrow u$. Likewise, if $q \in\, \downarrow_{\prec} u$ and $w \prec q$, then $w \in\, \downarrow_{\prec} u$. (Both follow from transitivity.)

**Explicit forms in $(Y, \preceq) = (\mathbb{R}^k, \leq)$.** With the product order, for $u \in \mathbb{R}^k$,

$$\downarrow u \;=\; \{\, z \in \mathbb{R}^k : z_i \leq u_i \; \forall i \,\} \;=\; u - \mathbb{R}^k_+,$$

$$\downarrow_{\prec} u \;=\; \{\, z \in \mathbb{R}^k : z_i \leq u_i \; \forall i, \; z \neq u \,\} \;=\; \big(u - \mathbb{R}^k_+\big) \setminus \{u\},$$

i.e., the lower $\leq$-orthant under $u$, with the corner point $u$ removed in the strict case.

**Intersection and coordinatewise meet.** For $u, v \in \mathbb{R}^k$, let $(u \wedge v)_i := \min\{u_i, v_i\}$. Then

$$\downarrow u \cap \downarrow v \;=\; \downarrow (u \wedge v).$$

If $u \parallel v$ (strict incomparability), then $u \wedge v \neq u$ and $u \wedge v \neq v$, hence

$$\downarrow_{\prec} u \cap \downarrow_{\prec} v \;=\; \{\, z : z \leq u \wedge v \,\},$$

so the strict down-sets still overlap (they coincide on the orthant under $u \wedge v$).

**Connection to nondominated sorting.** For a finite $S \subset \mathbb{R}^k$, an element $x \in S$ is *minimal* (Pareto-nondominated) iff

$$\neg \exists\, y \in S \text{ with } y \prec x \iff \big(S \setminus \{x\}\big) \cap \downarrow_{\prec} x \;=\; \varnothing.$$

Thus the first nondominated layer is $L_1 = Min(S) := \{x \in S : \; (S \setminus \{x\}) \cap \downarrow_{\prec} x = \emptyset\}$. Removing it and iterating constructs subsequent layers:

$$L_{t+1} \;=\; Min\Big(S \setminus \bigcup_{i=1}^{t} L_i\Big), \qquad t \geq 1.$$

(Using $\prec$ or $\preceq$ here is equivalent on finite $S$: "no strict dominator" $\Leftrightarrow$ "minimal in $\preceq$.") This perspective provides algebraic shortcuts for nondominated sorting, which we exploit later.

## 3 A propagation property of the strict Pareto order

In this section, we prove a propagation property of the strict product (Pareto) order on $\mathbb{R}^k$ and its symmetric counterpart. The result is established via concise proofs—algebraic and geometric (via strict down-sets)—and is visualized in the $k = 2$ case. We also derive an immediate corollary that lays the foundation for the algorithmic consequences developed later.

**Theorem 1** (Propagation under strict Pareto dominance). *Let $u, v, q \in \mathbb{R}^k$ and assume $u \parallel v$.*

*(a) If $q \prec v$, then $u \not\prec q$. Consequently (since $q = u$ would imply $u \prec v$), either $q \prec u$ or $q \parallel u$.*

*(b) Symmetrically, if $q \prec u$, then $v \not\prec q$. Consequently, either $q \prec v$ or $q \parallel v$.*

*One-line proof via transitivity.* Pareto dominance is transitive: if $x \prec y$ and $y \prec z$, then $x_i \leq y_i \leq z_i$ for all $i$ and $x \neq z$, hence $x \prec z$. Thus, if $q \prec v$ and $u \prec q$ both held, we would get $u \prec v$, contradicting $u \parallel v$. $\square$

*Algebraic proof.* Since $u \parallel v$, there exist indices $n, m$ with $v_n < u_n$ and $u_m < v_m$. Suppose $q \prec v$. Then $q_i \leq v_i$ for all $i$, so in particular $q_n \leq v_n < u_n$. If $u \prec q$ held, we would need $u_i \leq q_i$ for all $i$, which is impossible at $i = n$. Hence $u \not\prec q$. Equality $q = u$ would force $u \prec v$, contradicting $u \parallel v$ thus, the remaining possibilities are $q \prec u$ (which can happen if, e.g., $q_m \leq u_m$) or strict-incomparability $q \parallel u$ (if, e.g., $q_m > u_m$). The symmetric statement follows by exchanging $u$ and $v$. $\square$

**Remark 1.** *(i) The assumption $u \parallel v$ rules out equality and guarantees "crossing coordinates" (some coordinate where $u$ is smaller and another where $v$ is smaller).*

*(ii) The conclusion uses only strict relations: we prove $u \not\prec q$ (not the stronger $u \not\preceq q$), which is the sharp statement when one insists on strict dominance throughout.*

**Corollary 1** (No two-step bridge between incomparable anchors $u, v$). *If $v \parallel u$, the chains $u \prec q \prec v$ and $v \prec q \prec u$ are impossible.*

*Proof.* Immediate from transitivity of $\prec$: $u \prec q \prec v$ would imply $u \prec v$, contradicting $v \parallel u$. $\square$

*Geometric proof via strict down-sets.* Let $\downarrow_\prec x := \{z \in \mathbb{R}^k : z \prec x\}$ denote the strict down-set of $x$. We use two elementary facts (both immediate from the definition and transitivity of $\prec$):

(i) If $x \prec y$, then $\downarrow_\prec x \subset \downarrow_\prec y$ and $x \in \downarrow_\prec y$.

(ii) If $x \parallel y$, then neither $\downarrow_\prec x \subset \downarrow_\prec y$ nor $\downarrow_\prec y \subset \downarrow_\prec x$.

Assume $u \parallel v$ and $q \prec v$. Then by (i), $\downarrow_\prec q \subset \downarrow_\prec v$. If, for contradiction, $u \prec q$, then again by (i),

$$\downarrow_\prec u \subset \downarrow_\prec q \subset \downarrow_\prec v \quad \text{and} \quad u \in \downarrow_\prec q \subset \downarrow_\prec v.$$

The latter inclusion is exactly $u \prec v$, contradicting $u \parallel v$. Hence $u \not\prec q$. This proves (a). Part (b) is identical with $u$ and $v$ interchanged. $\square$

**Remark 2** (Contrapositive and closure view). *The contrapositive of the corollary is: if $u \prec q \prec v$, then $v \not\parallel u$ (indeed $u \prec v$).*

*In "closure" terms with strict down-sets $\downarrow_\prec x := \{z : z \prec x\}$:*

$$q \prec v \Rightarrow \downarrow_\prec q \subset \downarrow_\prec v, \qquad u \prec q \Rightarrow \downarrow_\prec u \subset \downarrow_\prec q.$$

*Hence $u \prec q \prec v \Rightarrow \downarrow_\prec u \subset \downarrow_\prec v$. But $u \parallel v$ forbids either strict containment $\downarrow_\prec u \subset \downarrow_\prec v$ or $\downarrow_\prec v \subset \downarrow_\prec u$ (there are "crossing coordinates"), so $u \prec q$ cannot occur when $q \prec v$.*

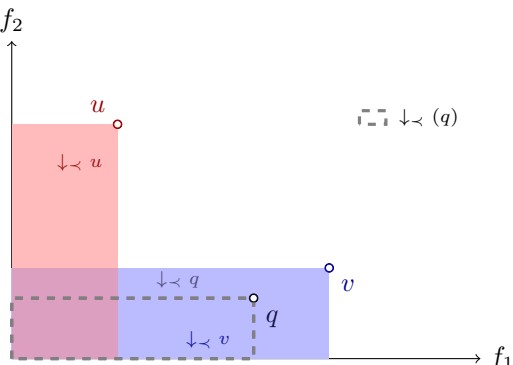

Figure 1: Strict down-sets (punctured $\leq$-orthants) for incomparable $u$ (red) and $v$ (blue). Given $q \prec v$, we have $\downarrow_\prec q \subset \downarrow_\prec v$. If $u \prec q$ also held, then $\downarrow_\prec u \subset \downarrow_\prec q \subset \downarrow_\prec v$ and $u \in \downarrow_\prec q \subset \downarrow_\prec v$, forcing $u \prec v$—contradiction.

## 4   Algorithmic consequence for Pareto archive insertion: a post-witness $u \prec q$ pruning rule

In this section, we define the notion of a *witness*, formalize the post-witness $u \prec q$ pruning implied by Theorem 1, and give a single-pass insertion algorithm for a mutually nondominated Pareto archive that implements this rule. We then prove the algorithm's correctness, quantify its comparison-level effect, discuss scan-order considerations, and give a simple random-input estimate that helps interpret when pruning is likely to be substantial.

**Terminology (witness).**   Let $S \subseteq \mathbb{R}^k$ be mutually nondominated and let $q \in \mathbb{R}^k$. Any $w \in S$ satisfying $q \prec w$ is called a *witness* for $q$ in $S$; the set of all witnesses is

$$W(q; S) := \{\, w \in S : \ q \prec w \,\}.$$

When scanning $S$ in some order, the *first witness* is the first element of $W(q; S)$ encountered by the scan, if such an element exists.

**Corollary 2** (Post-witness $u \prec q$ pruning rule for archive insertion). *Let $S \subseteq \mathbb{R}^k$ be mutually nondominated, and let $q \in \mathbb{R}^k$. Suppose there exists $v \in S$ with $q \prec v$. Then, for every $u \in S$ such that $u \parallel v$, one has $u \nprec q$. Consequently, all dominance checks of the form "$u \prec q$?" for such $u$ may be safely skipped.*

*Proof.* Immediate from Theorem 1(a) with the triple $(u, v, q)$. $\qquad\square$

**Corollary 3** (Archive case: global pruning after the first witness). *Let $S \subseteq \mathbb{R}^k$ be mutually nondominated, and fix any scan order of $S$. If the scan discovers a witness $w \in S$ with $q \prec w$, then every later-scanned element $u \in S$ satisfies $u \nprec q$. Therefore, after the first witness is found, all subsequent tests of the form "$u \prec q$?" may be omitted.*

*Proof.* Because $S \subseteq \mathbb{R}^k$ is mutually nondominated, every $u \in S \setminus \{w\}$ is incomparable with $w$. The conclusion then follows from Corollary 2. $\qquad\square$

**Application to incremental Pareto archive insertion.**   In incremental Pareto archive maintenance, inserting a candidate $q$ into a mutually nondominated archive $S$ typically consists of two tasks: first, reject $q$ if it is dominated by some archive element $a \in S$; second, if it is not rejected, remove any $a \in S$ such that $q \prec a$. The post-witness pruning rule prunes the first task *after* the first witness has been found: once some $w \in S$ satisfies $q \prec w$, Corollary 3 implies that no remaining $a \in S$ can dominate $q$. From that point onward, the scan need only test whether $q$ dominates each remaining element.

We describe insertion as a single pass over the archive $S$. The order in which the routine visits the elements $a \in S$ will be called the *scan order*. This may be the current in-memory order of the archive or any other chosen traversal order.

---

**Algorithm 1** Insert $q$ into a mutually nondominated archive $S$ using Corollary 2

---

**Require:** $S \subseteq \mathbb{R}^k$ is mutually nondominated; $q \in \mathbb{R}^k$
1: *witnessFound* $\leftarrow$ false
2: $R \leftarrow \emptyset$                       ▷ archive points already identified as dominated by $q$
3: **for all** $a \in S$ in the chosen scan order **do**
4:     **if** *witnessFound* = false **then** ▷ Before the first witness is found, both dominance directions are still possible
5:         **if** $a \prec q$ **then**
6:             **return** $S$                       ▷ $q$ is dominated by an archive point, so it is rejected
7:         **end if**
8:     **end if**
9:     **if** $q \prec a$ **then**
10:         $R \leftarrow R \cup \{a\}$
11:         *witnessFound* $\leftarrow$ true       ▷ Once a witness is found, later tests of the form $a \prec q$ are unnecessary
12:     **end if**
13: **end for**
14: **return** $(S \setminus R) \cup \{q\}$                       ▷ insert $q$ and remove all archive points dominated by it

---

**Theorem 2** (Correctness of Algorithm 1)**.** *Let $S \subseteq \mathbb{R}^k$ be mutually nondominated and let $q \in \mathbb{R}^k$.*

1. *If there exists $a \in S$ such that $a \prec q$, then Algorithm 1 rejects $q$ and returns $S$.*

2. *If no $a \in S$ satisfies $a \prec q$, then Algorithm 1 returns*

$$S' = (S \setminus R) \cup \{q\}, \qquad R = \{a \in S : q \prec a\},$$

*and $S'$ is mutually nondominated.*

*Proof.* Assume first that there exists $a^\star \in S$ with $a^\star \prec q$. We show that the algorithm returns $S$.

If no witness has been found before $a^\star$ is scanned, then at that step the algorithm still performs the test $a^\star \prec q$, which succeeds, and it immediately returns $S$.

If instead a witness $w \in S$ with $q \prec w$ had already been found before $a^\star$ was scanned, then $a^\star$ would be a later-scanned archive element. Since $S$ is mutually nondominated, one has $a^\star \parallel w$. Corollary 3 then implies that no later-scanned element can satisfy $u \prec q$, contradicting $a^\star \prec q$. Hence this case is impossible. Therefore, whenever some $a \in S$ satisfies $a \prec q$, the algorithm rejects $q$ and returns $S$.

Now assume that no $a \in S$ satisfies $a \prec q$. Then $q$ is not dominated by any archive element, so it should be accepted.

Before the first witness is found, the algorithm tests each scanned archive element in both directions, namely whether $a \prec q$ and whether $q \prec a$. Once the first witness $w$ with $q \prec w$ is found, Corollary 3 implies that no later-scanned archive element $u$ can satisfy $u \prec q$. Therefore all subsequent tests of the form "$u \prec q$?" may be omitted without risking rejection of a nondominated candidate. The algorithm continues to test whether $q \prec u$, and thus it collects exactly the set

$$R = \{a \in S : q \prec a\}.$$

Consequently, the returned set is

$$S' = (S \setminus R) \cup \{q\}.$$

It remains to show that $S'$ is mutually nondominated. By construction, every $a \in R$ is removed because $q \prec a$. Every survivor $s \in S \setminus R$ therefore satisfies $q \not\prec s$. By the standing assumption of this case, no element of $S$ satisfies $s \prec q$, hence in particular no survivor in $S \setminus R$ dominates $q$. Finally, the elements of $S \setminus R$ remain mutually nondominated because they are a subset of the mutually nondominated set $S$. Thus no two distinct elements of $S'$ dominate one another, so $S'$ is mutually nondominated. □

Theorem 2 establishes that Algorithm 1 is correct: it rejects $q$ exactly when $q$ is strictly dominated by an archive element, and otherwise returns the mutually nondominated update obtained by inserting $q$ and removing all archive points strictly dominated by it. The pruning rule therefore changes only the number of dominance tests performed during the scan, not the mathematical outcome of insertion. This makes it natural to quantify its effect first at the level of strict-dominance predicate calls.

**Remark 3** (Dominance-predicate complexity of insertion). *Let $m := |S|$ be the archive size at the start of insertion.*

*In the standard baseline linear-scan routine, an accepted insertion may require up to $2m$ strict-dominance predicate calls: for each $a \in S$, one call to test whether $a \prec q$ and one call to test whether $q \prec a$.*

*In Algorithm 1, consider an accepted insertion.*

*If a first witness is encountered at scan position $j \in \{1, \ldots, m\}$, then the first $j$ scanned elements (including the witness) require both tests, whereas after the first witness only tests of the form $q \prec u$ remain. Hence the total number of strict-dominance predicate calls is*

$$2j + (m - j) = m + j.$$

*Thus the post-witness savings equal*

$$2m - (m + j) = m - j$$

*predicate calls relative to the baseline accepted-insertion scan.*

*If no witness is found, then the insertion is still accepted, but every scanned archive element requires both tests, so the total is $2m$.*

*Accordingly, the worst case for an accepted insertion remains $2m$, while the best case is $m + 1$, attained when the first scanned element is already a witness. Therefore the pruning rule preserves the $O(m)$ worst-case asymptotic complexity of linear-scan archive insertion while improving the constant factor whenever a witness appears before the end of the scan.*

*If one counts each strict-dominance predicate call as unit cost, then over a sequence of $N$ insertions with archive sizes $m_1, \ldots, m_N$, the total worst-case number of predicate calls is*

$$O\left(\sum_{t=1}^{N} m_t\right),$$

*and since $m_t \leq t - 1$, this is $O(N^2)$. If one instead resolves each predicate by scanning all $k$ coordinates, the corresponding coordinate-level worst-case bound is $O(km)$ per insertion and $O(kN^2)$ over $N$ insertions.*

Remark 3 shows that the benefit of the pruning rule is governed by how early a first witness appears in the scan. This yields an exact comparison-count formula for accepted insertions, together with the induced best-case and worst-case bounds. We do not state an average-case complexity bound for archive insertion, because such a bound is inherently model-dependent: it depends not only on the distribution of incoming points, but also on the scan order, the evolving nondominance structure of the archive, and the dependencies induced by archive curation. Instead, Remark 3 gives an exact comparison-count formula conditional on the first-witness position, together with the induced best-case and worst-case bounds.

**Remark 4** (Extensions to weak and $\varepsilon$-dominance). *The post-witness pruning argument used above relies only on two ingredients: first, the transitivity of the comparison relation used for pruning; and second, incomparability being defined with respect to that same relation. It therefore extends beyond the exact strict*

*Pareto relation to other transitive dominance relations used in multiobjective optimization. Two common cases are as follows.*

*(a) Weak dominance. Suppose the algorithm uses weak dominance $\preceq$, with strict part $\prec$, and defines incomparability with respect to $\preceq$. Then the same post-witness pruning remains valid with a weak witness: if $q \preceq v$ and $v \parallel_{\preceq} u$, then $u \not\prec q$. Indeed, if $u \prec q$ also held, then $u \preceq q \preceq v$, hence $u \preceq v$, contradicting $v \parallel_{\preceq} u$.*

*(b) $\varepsilon$-dominance. Let $\prec_{\varepsilon}$ be a transitive $\varepsilon$-dominance relation, and let $\parallel_{\varepsilon}$ denote incomparability defined with respect to $\prec_{\varepsilon}$. Then the same propagation argument applies: if $q \prec_{\varepsilon} v$ and $v \parallel_{\varepsilon} u$, one cannot have $u \prec_{\varepsilon} q$. This includes, for example, the additive relation*

$$u \prec_{\varepsilon} v \iff u_i \leq v_i - \varepsilon_i \quad \text{for all } i,$$

*for which transitivity follows immediately from coordinatewise inequality.*

**Back-of-the-envelope proxy calculation under random inputs.** The following estimate is useful for gauging behavior on uncurated point sets generated by random sampling—for example, in early stages of processing a random point cloud—where pairwise comparabilities may still be common.

**Proposition 1** (Comparability baseline and proxy omitted fraction under i.i.d. continuous sampling). *Assume the following sampling model, used solely for estimation: points in $\mathbb{R}^k$ are sampled independently; for each coordinate $i \in \{1, \ldots, k\}$, the random variables are independent across $i$, identically distributed across points, and have continuous marginals. In particular, for independent scalar copies $U_i, V_i$ we have $\Pr(U_i = V_i) = 0$ and $\Pr(U_i < V_i) = \Pr(V_i < U_i) = \frac{1}{2}$.*

*Let $U, V \in \mathbb{R}^k$ be independent points drawn from this model. Then*

$$\Pr\big(U \text{ and } V \text{ are comparable under } \prec\big) = 2^{1-k}, \qquad \Pr(U \parallel V) = 1 - 2^{1-k}.$$

*If, heuristically, the first witness $w$ and each remaining archive element $u$ are treated as independent draws from the same model (thus ignoring the bias introduced by conditioning on $q \prec w$), then the expected fraction of the remaining "$\cdot \prec q$?" checks that can be omitted is $1 - 2^{1-k}$.*

*Proof.* Let $U = (U_1, \ldots, U_k)$ and $V = (V_1, \ldots, V_k)$ be independent $\mathbb{R}^k$–valued random vectors such that, for each $i$, $U_i$ and $V_i$ are i.i.d. with a continuous distribution and the coordinates are independent across $i$. Then $\Pr(U_i = V_i) = 0$ and, by exchangeability, $\Pr(U_i < V_i) = \Pr(V_i < U_i) = \frac{1}{2}$. Independence of the coordinates then gives

$$\Pr(U \leq V) = \Pr(U_1 \leq V_1, \ldots, U_k \leq V_k) = \prod_{i=1}^{k} \Pr(U_i \leq V_i) = 2^{-k}.$$

Because $\Pr(U = V) = 0$, we have $\Pr(U \prec V) = \Pr(U \leq V, U \neq V) = 2^{-k}$. By symmetry, $\Pr(V \prec U) = 2^{-k}$. The events $\{U \prec V\}$ and $\{V \prec U\}$ are disjoint, so

$$\Pr(U \text{ and } V \text{ comparable under } \prec) = \Pr(U \prec V) + \Pr(V \prec U) = 2^{1-k},$$

and therefore $\Pr(U \parallel V) = 1 - 2^{1-k}$.

For the post-witness pruning fraction, if one heuristically models the discovered witness $w$ as an independent generic draw and each remaining $u$ as an i.i.d. draw independent of $w$, then $\Pr(u \parallel w) = 1 - 2^{1-k}$ by the calculation above. Hence the expected fraction of omitted "$\cdot \prec q$?" checks is $1 - 2^{1-k}$. $\square$

**Remark 5** (Archives versus i.i.d. samples; scope of the proxy). *The estimate in Proposition 1 is intended only as a baseline for uncurated point sets sampled independently from a continuous distribution on $\mathbb{R}^k$. In a mutually nondominated archive $S$, once a witness $w$ with $q \prec w$ is found, Corollary 3 implies that every remaining $u \in S \setminus \{w\}$ is incomparable with $w$; hence all subsequent tests of the form "$u \prec q$?" may be omitted deterministically. If exact duplicates are permitted, then under strict dominance duplicates remain incomparable with $w$ and cannot dominate $q$; under weak dominance they may instead require separate*

*equality handling. In either case, the pruning rule remains sound. Real archives generally violate the i.i.d. assumptions: nondominance filtering induces dependencies across points, and optimization dynamics may introduce correlations across both points and coordinates. Consequently, the i.i.d. comparability probability $2^{1-k}$ should not be used to predict pruning rates in curated archives; it serves only as a typical-case heuristic for genuinely i.i.d. samples.*

**Practical integration and scan order.** The post-witness $u \prec q$ pruning rule is an insertion-time pruning primitive: it identifies dominance tests of the form "$u \prec q$?" that become redundant after a witness has been encountered during the scan. Its correctness is independent of the scan order. However, the scan order affects how early a witness is found, and hence how many remaining "$u \prec q$?" checks can be avoided. Earlier witnesses yield larger practical savings.

Accordingly, the rule can be combined with any traversal policy or archive data structure that tends to expose witnesses early, without altering the correctness guarantees proved above. The role of scan-order heuristics is therefore not to justify the rule, but to increase the frequency and timing of the situations in which the rule becomes active.

### 4.1  Empirical illustration: dominance-comparison savings

To illustrate the archive-level pruning effect of Algorithm 1, we include a small synthetic experiment. Its purpose is explanatory rather than evidentiary: correctness is already established in Theorems 1 and 2, and the experiment is not intended as a numerical validation of Proposition 1, whose calculation concerns an uncurated i.i.d. comparison model rather than a dynamically maintained mutually nondominated archive. Instead, the results below make the practical consequences of the post-witness pruning rule directly observable under a controlled synthetic insertion process, in particular the relation between witness occurrence, first-witness position, and the resulting reduction in dominance-predicate calls.

**Setup.** We work directly in objective space and treat each sampled point as an objective vector in $\mathbb{R}^k$ for a minimization problem. For each pair $(N, k)$, we draw $N$ i.i.d. points uniformly from $[0, 1]^k$ and insert them sequentially into an initially empty archive. We compare two insertion procedures applied to the same input sequence and the same scan order: (i) a standard incremental archive-insertion routine (baseline), and (ii) the same routine augmented with the post-witness $u \prec q$ pruning rule. In both cases, the archive is maintained as mutually nondominated under strict Pareto dominance. Cost is measured by the total number of calls to the strict-dominance predicate over the full insertion run; each such call compares two vectors under strict Pareto dominance and may inspect up to $k$ coordinates. We repeat each configuration for $R = 20$ independent runs and report empirical means.

Accordingly, the empirical reductions reported here should be interpreted relative to the deterministic post-witness archive rule, not as estimates of the quantity $1 - 2^{1-k}$ from Proposition 1.

**Primary comparison metric and diagnostics.** Let $T_{\text{baseline}}$ and $T_{\text{pruned}}$ denote the total numbers of strict-dominance predicate calls made by the baseline and pruned insertion routines, respectively, over one full insertion run. We report the relative reduction

$$\text{Red} := 1 - \frac{T_{\text{pruned}}}{T_{\text{baseline}}},$$

displayed in the tables as the percentage $100\,\text{Red}$.

In addition to this overall reduction, we report two mechanism-level diagnostics computed during insertion. At a given insertion step, let $S$ denote the archive at entry to the insertion routine and let $q$ denote the current candidate. A *witness event* occurs if $q$ strictly dominates at least one current archive element, i.e., if there exists $w \in S$ such that $q \prec w$. We report:

- *Witness rate:* the fraction of insertions at which a witness event occurs (i.e., $\exists\, w \in S$ with $q \prec w$).

- *First-witness position (First-pos):* conditional on a witness event, if $i \in \{1, \ldots, |S|\}$ is the scan index of the first encountered witness, we report the normalized position $i/|S| \in (0, 1]$.

These diagnostics quantify (i) how often the pruning rule is triggered and (ii) whether it is triggered early enough to prune many remaining "$u \prec q$?" checks.

**Insertion orders.** We report two insertion orders for the same sampled point set. The first is *random order*, namely the sampled order itself. The second is a deterministic *witness-rich order*, constructed as follows. If the sampled outcome vectors are $z^{(1)}, \ldots, z^{(N)} \in [0, 1]^k$, define the scalar score

$$s(z) := \sum_{i=1}^{k} z_i.$$

The witness-rich order is obtained by sorting the sampled points in decreasing order of $s(z)$, i.e., from larger total objective value to smaller total objective value. Thus, relatively poorer outcomes are inserted earlier and relatively better outcomes later, increasing the likelihood that an incoming point dominates at least one current archive element and therefore produces a witness event. Throughout, the same insertion order is used for both the baseline and pruned routines within each run.[5].

**Results and interpretation.** Figure 2 reports the mean reduction in dominance-predicate calls as a function of $k$ for $N \in \{500, 1000, 2000\}$, under both insertion orders. Figure 4.1 reports the corresponding witness-rate diagnostic, while Tables 1–6 report the underlying numerical values for reduction, witness rate, and first-witness position.

**Numeric reporting.** All table entries are means over $R = 20$ independent runs. Reduction is reported as a percentage with one decimal place; witness rate and First-pos are reported with three decimal places; and mean dominance-predicate call counts are reported with one decimal place. Values are rounded for presentation.

Across all tested settings, the observed savings track the activation mechanism of the rule: reductions are small when witness events are rare and/or discovered late in the scan, and substantially larger when witness events occur frequently and early. This is exactly what the post-witness pruning rule predicts, since it becomes active only after the first witness is found and then removes the remaining "$u \prec q$?" checks.

---

[5]This reordering is used only to illustrate the pruning mechanism in a regime where witnesses are frequent; it is not intended as a model of typical optimizer-generated streams

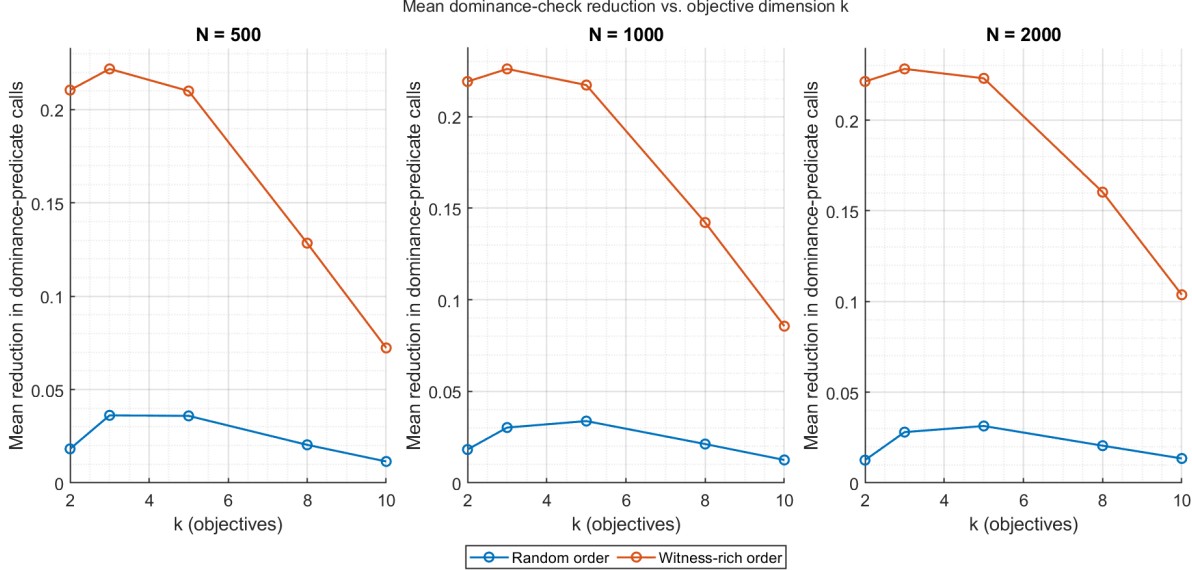

Figure 2: Mean reduction in dominance-predicate calls achieved by the post-witness pruning rule relative to the baseline insertion routine, as a function of the number of objectives $k$, for $N \in \{500, 1000, 2000\}$ under random order and witness-rich order. Reduction is defined as $1 - T_{\mathrm{pruned}}/T_{\mathrm{baseline}}$, and values are averaged over $R = 20$ independent runs.

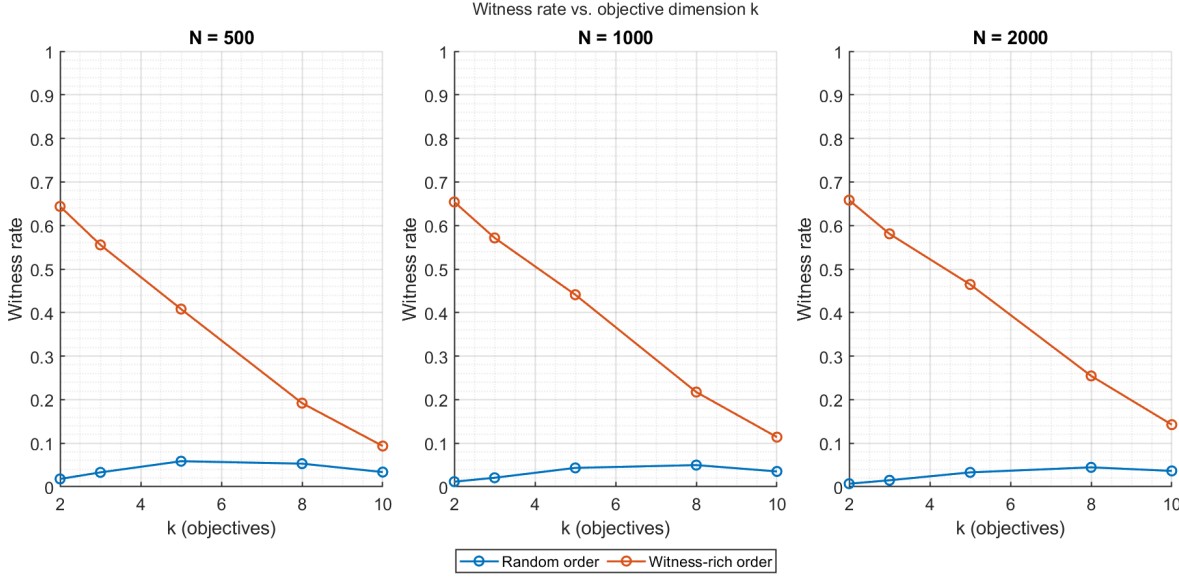

Figure 3: Witness rate versus $k$ for $N \in \{500, 1000, 2000\}$ under random order and witness-rich order. Values are averaged over $R = 20$ independent runs. First-witness-position values are reported in Tables 1–6.

Table 1: Empirical diagnostics for $N = 500$ (means over $R = 20$ runs). Reduction is $1 - T_{\mathrm{pruned}}/T_{\mathrm{baseline}}$.

| | random order | | | witness-rich order | | |
|---|---|---|---|---|---|---|
| $k$ | Red. (%) | Witness-rate | First-pos | Red. (%) | Witness-rate | First-pos |
| 2 | 1.8 | 0.018 | 0.551 | 21.0 | 0.644 | 0.372 |
| 3 | 3.6 | 0.033 | 0.457 | 22.2 | 0.555 | 0.244 |
| 5 | 3.6 | 0.059 | 0.394 | 21.0 | 0.408 | 0.132 |
| 8 | 2.0 | 0.053 | 0.415 | 12.8 | 0.192 | 0.084 |
| 10 | 1.1 | 0.034 | 0.439 | 7.2 | 0.093 | 0.057 |

Table 2: Mean total dominance-predicate calls for $N = 500$.

| | random order | | witness-rich order | |
|---|---|---|---|---|
| $k$ | Baseline | Pruned | Baseline | Pruned |
| 2 | 1123.2 | 1102.8 | 20102.0 | 15873.6 |
| 3 | 4119.1 | 3965.8 | 62227.7 | 48434.3 |
| 5 | 33758.3 | 32504.6 | 150665.2 | 119033.8 |
| 8 | 137200.5 | 134385.8 | 223853.1 | 195113.9 |
| 10 | 201934.8 | 199615.0 | 241747.5 | 224269.5 |

### 4.2 Noisy/stochastic evaluations and certified dominance

The main results of this paper establish inference-based pruning rules in a deterministic setting, where each evaluated outcome vector $F(x) \in \mathbb{R}^k$ is fixed and a pairwise Pareto-order comparison—e.g., testing whether $F(x) \preceq F(y)$—is therefore an exact statement. In many machine-learning applications, however, objective values are not computed exactly but are instead estimated from subsamples of data; for example, each objective may be an empirical risk evaluated on a random mini-batch $B$ of a dataset, yielding an estimated outcome vector

$$\widehat{F}_B(x) \;=\; (\widehat{f}_{1,B}(x), \ldots, \widehat{f}_{k,B}(x)) \in \mathbb{R}^k, \qquad x \in \mathcal{F},$$

Because $\widehat{F}_B(x)$ depends on a random $B$, the truth value of a comparison such as $\widehat{F}_B(x) \preceq \widehat{F}_B(y)$ is itself random, so the primitive comparison operation underlying the pruning rule becomes stochastic. The purpose of this subsection is therefore to make this shift from exact to stochastic comparisons explicit and to introduce a conservative, certification-based comparison relation under which the same order-theoretic pruning logic can still be applied safely.

**Lemma 1** (Stochasticity of order tests). *Fix $x, y \in \mathcal{F}$ and let $B$ denote a random mini-batch. Define*

$$\mathsf{Test}_B(x, y) \;:=\; \mathbf{1}\!\left\{\widehat{F}_B(x) \preceq \widehat{F}_B(y)\right\} \in \{0, 1\},$$

*where $\mathbf{1}\{\cdot\}$ denotes the indicator of an event. Assume that, for fixed $x$ and $y$, the computed estimates $\widehat{F}_B(x)$ and $\widehat{F}_B(y)$ are well-defined functions of the batch $B$ (as is the case for standard mini-batch empirical-risk estimators). Then $\mathsf{Test}_B(x, y)$ is a random variable (equivalently, the pairwise order test is stochastic).*

*Proof.* By definition,

$$\{\mathsf{Test}_B(x, y) = 1\} \;=\; \{\widehat{F}_B(x) \preceq \widehat{F}_B(y)\}, \qquad \{\mathsf{Test}_B(x, y) = 0\} \;=\; \{\widehat{F}_B(x) \npreceq \widehat{F}_B(y)\}.$$

Once a realization of $B$ is given, the evaluation rule deterministically produces $\widehat{F}_B(x)$ and $\widehat{F}_B(y)$, and the relation $\preceq$ on $\mathbb{R}^k$ is fixed; hence each of the above statements is unambiguously true or false for that realized batch. Therefore $\{\mathsf{Test}_B(x, y) = 1\}$ and $\{\mathsf{Test}_B(x, y) = 0\}$ are events determined by the random draw of $B$, and $\mathsf{Test}_B(x, y)$ is a random variable. Equivalently, these events are the preimages of $\{1\}$ and $\{0\}$ under the mapping $B \mapsto \mathsf{Test}_B(x, y)$. □

Table 3: Empirical diagnostics for $N = 1000$ (means over $R$ runs). Reduction is $1 - T_{\text{pruned}}/T_{\text{baseline}}$.

| | random order | | | witness-rich order | | |
|---|---|---|---|---|---|---|
| $k$ | Red. (%) | Witness-rate | First-pos | Red. (%) | Witness-rate | First-pos |
| 2 | 1.8 | 0.012 | 0.527 | 21.9 | 0.654 | 0.350 |
| 3 | 3.0 | 0.021 | 0.446 | 22.6 | 0.571 | 0.239 |
| 5 | 3.4 | 0.043 | 0.389 | 21.7 | 0.441 | 0.140 |
| 8 | 2.1 | 0.050 | 0.392 | 14.2 | 0.217 | 0.077 |
| 10 | 1.3 | 0.035 | 0.421 | 8.6 | 0.114 | 0.057 |

Table 4: Mean total dominance-predicate calls for $N = 1000$.

| | random order | | witness-rich order | |
|---|---|---|---|---|
| $k$ | Baseline | Pruned | Baseline | Pruned |
| 2 | 1816.4 | 1781.8 | 55940.4 | 43681.7 |
| 3 | 7367.7 | 7139.4 | 192066.2 | 148638.5 |
| 5 | 73157.4 | 70642.9 | 537564.7 | 420773.5 |
| 8 | 447027.3 | 437420.5 | 870754.3 | 746925.6 |
| 10 | 712744.4 | 703774.2 | 955286.9 | 873529.2 |

**Remark 6.** *The use of the weak order $\preceq$ in Lemma 1 is only illustrative: the same conclusion holds for strict dominance or any other deterministic comparison predicate built from the estimated outcomes $\widehat{F}_B(x)$ and $\widehat{F}_B(y)$. The lemma is used only to formalize that mini-batch-based comparison outcomes are random. The actual pruning logic developed below is then formulated with the certified strict relation $\prec_{\text{cert}}$, since the post-witness pruning argument in this paper relies on a strict transitive relation together with incomparability defined relative to that same relation.*

Lemma 1 shows that, when objective values are estimated from random mini-batches, the primitive order test used to trigger post-witness pruning is no longer an exact deterministic statement. Since archive insertion and the post-witness pruning rule use such comparisons as premises for further inferences, the next question is therefore: *under what conditions can a stochastic comparison outcome be promoted to a trusted relation on which pruning and comparison-test skips may safely rely?* One convenient approach is to introduce a conservative *certified* dominance relation derived from uncertainty bounds and then use this certified relation consistently for witnesses, tests, and incomparability, so that the order-theoretic pruning argument applies without modification.

**Certified dominance (conservative pruning under uncertainty).** To preserve the correctness of the post-witness pruning rule (Corollaries 2 and 3) in the presence of stochastic objective evaluations, we apply pruning only when a dominance relation can be *certified* from uncertainty bounds on the underlying objective values. Concretely, suppose that whenever two evaluated decisions $x, y \in \mathcal{F}$ are to be compared, one can form random bounds $L_i(x) \leq U_i(x)$ and $L_i(y) \leq U_i(y)$ such that, with probability at least $1 - \alpha$, the joint coverage event

$$\mathcal{E}(x, y) := \bigcap_{i=1}^{k} \Big( f_i(x) \in [L_i(x), U_i(x)] \ \wedge \ f_i(y) \in [L_i(y), U_i(y)] \Big)$$

holds.

(For example, in empirical-risk settings, such bounds can be obtained by repeated mini-batch evaluations together with standard concentration inequalities, yielding simultaneous confidence bounds across objectives at level $1 - \alpha$.)

Table 5: Empirical diagnostics for $N = 2000$ (means over $R$ runs). Reduction is $1 - T_{\text{pruned}}/T_{\text{baseline}}$.

| | random order | | | witness-rich order | | |
|---|---|---|---|---|---|---|
| $k$ | Red. (%) | Witness-rate | First-pos | Red. (%) | Witness-rate | First-pos |
| 2 | 1.3 | 0.007 | 0.504 | 22.1 | 0.658 | 0.343 |
| 3 | 2.8 | 0.015 | 0.417 | 22.8 | 0.581 | 0.240 |
| 5 | 3.1 | 0.033 | 0.378 | 22.3 | 0.465 | 0.141 |
| 8 | 2.1 | 0.045 | 0.394 | 16.0 | 0.255 | 0.078 |
| 10 | 1.3 | 0.036 | 0.403 | 10.4 | 0.143 | 0.056 |

Table 6: Mean total dominance-predicate calls for $N = 2000$.

| | random order | | witness-rich order | |
|---|---|---|---|---|
| $k$ | Baseline | Pruned | Baseline | Pruned |
| 2 | 4127.9 | 4076.2 | 156281.1 | 121697.8 |
| 3 | 14850.3 | 14420.1 | 610286.4 | 471000.5 |
| 5 | 166768.6 | 161384.6 | 1904410.6 | 1479707.8 |
| 8 | 1328793.2 | 1301376.1 | 3332871.7 | 2798696.4 |
| 10 | 2425316.5 | 2392552.0 | 3748177.0 | 3359558.0 |

Define the *certified strict dominance* relation on decisions by

$$x \prec_{\text{cert}} y \iff \big(\forall i,\ U_i(x) < L_i(y)\big),$$

and define certified incomparability $\|_{\text{cert}}$ with respect to $\prec_{\text{cert}}$ in the usual way.

**Lemma 2** (Soundness on the coverage event). *Assume the pairwise coverage event $\mathcal{E}(x,y)$ holds. If $x \prec_{\text{cert}} y$, then $f(x) \prec f(y)$.*

*Proof.* If $x \prec_{\text{cert}} y$, then $U_i(x) < L_i(y)$ for every $i$. On $\mathcal{E}(x,y)$ we have $f_i(x) \leq U_i(x)$ and $L_i(y) \leq f_i(y)$, hence $f_i(x) < f_i(y)$ for all $i$, i.e. $f(x) \prec f(y)$. $\square$

In particular, on the coverage event $\mathcal{E}$ the certified relation $\prec_{\text{cert}}$ is a conservative proxy for the true strict Pareto order: whenever certification declares $x \prec_{\text{cert}} y$, the corresponding true outcomes satisfy $F(x) \prec F(y)$. Thus, if the pruning logic is applied using $\prec_{\text{cert}}$ in place of $\prec$, it remains logically sound and—on $\mathcal{E}$—does not introduce false dominance conclusions about the underlying objectives.

Thus, the role of certification is twofold: it replaces raw stochastic comparisons by a conservative strict relation defined from uncertainty bounds, and, on the coverage event, it guarantees that this relation is a conservative approximation of the true Pareto order.

**Propagation/pruning under the certified relation.** The propagation/pruning argument used in this paper is order-theoretic: it relies only on (i) transitivity of the strict relation and (ii) defining incomparability with respect to the same strict relation used in the tests. We verify these properties for $\prec_{\text{cert}}$.

**Lemma 3** (Transitivity of $\prec_{\text{cert}}$). *If $x \prec_{\text{cert}} y$ and $y \prec_{\text{cert}} z$, then $x \prec_{\text{cert}} z$.*

*Proof.* Fix $i \in \{1, \ldots, k\}$. From $x \prec_{\text{cert}} y$ we have $U_i(x) < L_i(y)$, and from $y \prec_{\text{cert}} z$ we have $U_i(y) < L_i(z)$. Since $L_i(y) \leq U_i(y)$, it follows that

$$U_i(x)\ <\ L_i(y)\ \leq\ U_i(y)\ <\ L_i(z),$$

hence $U_i(x) < L_i(z)$. This holds for all $i$, so $x \prec_{\text{cert}} z$. $\square$

We now revert to the witness notation used in the main pruning rule (cf. Corollaries 2 and 3): $q \in \mathcal{F}$ denotes the incoming evaluated decision, $w \in \mathcal{F}$ a certified witness for $q$ (i.e., $q \prec_{\mathrm{cert}} w$), and $u \in \mathcal{F}$ a remaining archive element.

**Proposition 2** (Certified witness implies safe skips). *If $q \prec_{\mathrm{cert}} w$ and $w \parallel_{\mathrm{cert}} u$, then one cannot have $u \prec_{\mathrm{cert}} q$. Equivalently, any skip decision justified by a certified witness cannot suppress a later $\prec_{\mathrm{cert}}$-dominator.*

*Proof.* Assume for contradiction that $u \prec_{\mathrm{cert}} q$ and $q \prec_{\mathrm{cert}} w$. By Lemma 3, this implies $u \prec_{\mathrm{cert}} w$, contradicting $w \parallel_{\mathrm{cert}} u$. □

**Preventing catastrophic archive collapse and correcting noise-induced deletions.** A natural algorithmic concern in the mini-batch evaluation regime is whether a single incoming evaluated decision $q \in \mathcal{F}$, assessed using one random mini-batch $B$, could be spuriously judged so favorable that it appears to dominate many—or even all—elements of the current archive, thereby triggering a cascade of deletions that empties the archive, even though this apparent superiority is merely a sampling artifact. In the framework developed here, that concern must be analyzed with respect to the relation that actually governs deletion. Archive elements are not removed on the basis of a raw batch-wise comparison such as $\widehat{F}_B(q) \preceq \widehat{F}_B(u)$; rather, removal is permitted only when the stronger certified relation $q \prec_{\mathrm{cert}} u$ holds, meaning that the uncertainty bounds separate in the dominance direction across all objectives.

This yields an immediate *abstention principle*: if dominance is not certified—for example, because the uncertainty bounds overlap in at least one objective—then the algorithm neither prunes nor skips on the basis of that comparison. More concretely, under the certified-update policy an archive element $u$ is removed by an incoming candidate $q$ only if $q \prec_{\mathrm{cert}} u$, that is, only if the uncertainty bounds for $q$ and $u$ separate in the dominance direction in every objective. Consequently, a single mini-batch evaluation that makes $q$ appear spuriously optimistic cannot by itself eliminate the current archive unless this certification condition is satisfied simultaneously against every retained element. Thus, catastrophic archive collapse is not triggered by an isolated noisy comparison as such, but only by comparisons that pass the chosen certification rule; moreover, on the corresponding joint coverage event for the comparisons involved, those certified prunings are consistent with true Pareto dominance. Formally, this conclusion is exact on the simultaneous coverage event for all certified comparisons invoked by the update; if only pairwise coverage bounds are available, a familywise guarantee over one insertion step requires an additional union bound or a simultaneous confidence construction.

A second concern is how to recover from erroneous prunings if deletions are performed on the basis of non-certified, and therefore potentially noisy, comparisons—for example, to reduce per-iteration comparison cost. In that setting, some removed archive elements may later turn out not to have been truly dominated, so an explicit correction mechanism is needed. Two simple safeguards are natural. The first is *reversible deletion*: instead of discarding a removed element permanently, one places it in a shadow buffer for a fixed horizon and reinstates it if subsequent higher-fidelity evaluations or repeated comparisons contradict the earlier deletion. The second is *periodic rebuild*: one recomputes the archive from the accumulated pool of evaluated decisions by re-extracting its minimal elements under the current, possibly refined, comparison rule. Both mechanisms ensure that deletions induced by noisy comparisons need not be irreversible.

The certified-bounds construction above controls stochastic evaluations by replacing raw comparisons with a conservative relation supported by uncertainty bounds. When such bounds are unavailable, inconvenient to maintain, or unnecessarily elaborate for the application at hand, one can instead work with a simpler abstraction that models the comparison rule itself as noisy and controls its reliability by repetition.

**Noisy-comparator abstraction (repeated confirmations).** As an alternative to explicit confidence bounds, suppose that each invocation of a comparison used to certify the witness relation has probability at most $\delta$ of falsely returning the label needed to declare that witness, and that repeated evaluations of the same comparison are independent. If a skip decision is made only after obtaining $r$ independent confirmations of the same witness relation, then the probability that the skip is triggered by a false witness is at most $\delta^r$.

Consequently, choosing

$$r \geq \left\lceil \frac{\log(\alpha)}{\log(\delta)} \right\rceil, \qquad \alpha, \delta \in (0, 1),$$

ensures that the per-skip error probability is at most $\alpha$.

This guarantee is local and probabilistic rather than deterministic: it controls the probability of an incorrect skip at a single pruning event under the stated independence and error-rate assumptions. Extending it to a guarantee over an entire insertion step or a full run requires an additional union bound or a more refined dependence analysis. In empirical-risk settings, repeated confirmations may be implemented by reevaluating the same comparison on independently drawn mini-batches; increasing the batch size provides a complementary mechanism for reducing the single-comparison error level $\delta$ itself.

## 5    Discussion

**The main result.**    At an abstract order-theoretic level, the propagation property established in this paper is a consequence of the transitivity of the strict Pareto dominance relation $\prec$ on $\mathbb{R}^k$—that is, of the strict part of the underlying coordinatewise product order $\preceq$—together with strict incomparability defined relative to that same relation. The term *propagation* refers here to the propagation of a *constraint* on which further strict dominance relations remain possible once part of the local order structure is known. More precisely, once two anchors $u$ and $v$ are fixed as strictly incomparable, knowledge of one strict dominance relation involving a third point $q$ immediately excludes another: for example, $q \prec v$ forces $u \nprec q$, and symmetrically $q \prec u$ forces $v \nprec q$. In this sense, what propagates is negative information: a restriction on the admissible two-step strict-dominance configurations around an incomparable pair.

**What is new, and what is not.**    The propagation property established here should not be confused with the transitivity of $\prec$ itself. Transitivity is an underlying order-theoretic principle; the propagation property is a more specific derived statement obtained when transitivity is read together with strict incomparability defined relative to that same relation. Thus, the logical ingredients are classical, and the paper does not claim a new foundational law of Pareto order. The novelty lies elsewhere: in explicitly isolating this particular consequence as a distinct statement in its own right, recognizing that it yields a reusable pruning rule, and showing that this rule has direct algorithmic force in Pareto-order processing in dominance-testing routines. In the archive-insertion setting, that force becomes concrete: once a witness $w$ with $q \prec w$ has been found, every subsequent test of the form "$u \prec q$?" is logically impossible under the archive invariant and may therefore be omitted without changing the returned updated archive. The contribution of the paper is therefore the precise synthesis, formulation, and exploitation of a consequence of known ingredients that, in this explicit form, appears not to have been previously isolated in the multiobjective-optimization literature.

**Curated archives versus general finite sets.**    The strongest algorithmic consequence arises in the setting of a curated Pareto archive, that is, a finite set $S \subset \mathbb{R}^k$ that is mutually nondominated under $\prec$. In that setting, once a witness $w \in S$ with $q \prec w$ is found during insertion of a candidate $q$, every remaining archive element $u \in S \setminus \{w\}$ is automatically incomparable with $w$ by the archive invariant. The propagation property then yields $u \nprec q$ for every such $u$. Hence, after the first witness is encountered, every remaining test of the form "$u \prec q$?" is logically redundant. This is not heuristic, approximate, or average-case; it is a deterministic consequence of the archive invariant.

The situation is different for a general finite set that has not yet been filtered to nondominance. In such a set, a remaining point may still be comparable with the witness, so one cannot conclude that all later "$u \prec q$?" checks are unnecessary. What survives in that broader setting is the local rule of Corollary 2: after a witness $w$ with $q \prec w$ is found, one may skip "$u \prec q$?" only for those remaining points $u$ that satisfy $u \parallel w$. Thus, the same order-theoretic principle underlies both settings, but the strongest pruning rule depends on the maintained invariant of mutual nondominance. Whether this local rule yields net computational savings in a non-curated set depends additionally on the cost of establishing $u \parallel w$ relative to the skipped comparison.

**Correctness versus realized savings.**    It is of value to distinguish sharply between correctness and realized savings. Correctness is purely logical: once the assumptions of the corresponding corollary are

satisfied, the omitted dominance tests cannot change the returned archive. Realized savings, however, depend on *when* a witness is encountered in the scan. If a witness is found early, many remaining "$u \prec q$?" tests are available to be eliminated; if it is found late, the same rule remains valid but removes fewer tests. Scan order therefore affects efficiency but not correctness.

This also clarifies the meaning of witness-first strategies. Their role is not to make the pruning rule valid; validity follows entirely from the propagation property together with the maintained archive invariant. Their role is to increase the chance that a witness, if one exists, is encountered early enough to yield substantial pruning. In that precise sense, witness-oriented traversal is an efficiency heuristic layered on top of a correctness-preserving logical rule.

**What the empirical section does, and does not, show.** The empirical illustration in Section 4.1 was included to make the comparison-saving mechanism of the pruning rule visible, not to validate the theorem. The theorem and the archive-insertion rule are already proved formally in Theorems 1 and 2. The empirical section serves two narrower purposes. First, it quantifies the reduction in dominance-predicate calls in controlled synthetic streams. Second, it reports two diagnostics that explain those reductions: the *witness rate*, which measures how often witness events occur, and the *first-witness position*, which measures how early the first witness is encountered when one exists. These diagnostics are not ancillary; they are the empirical quantities most directly connected to the way the pruning rule generates savings. The empirical section should therefore be read as a synthetic diagnostic illustration of the pruning mechanism, not as a full benchmark of complete optimization pipelines or modern end-to-end nondominated-sorting implementations.

The results are consistent with this interpretation. When witness events are rare, or when they tend to occur late in the scan, the reduction in total dominance-predicate calls is modest. When witness events are frequent and are typically encountered early, the reduction is substantially larger. This is exactly what the post-witness pruning rule predicts: the rule saves work only after a witness appears, and the amount saved is determined by how much of the scan remains after that point.

**Proxy-model analysis versus archive logic.** The i.i.d. estimate in Proposition 1 and the deterministic archive result play different roles and should not be conflated. The archive result states what is always true once a witness has been found in a mutually nondominated archive: all subsequent checks of the form "$u \prec q$?" may be omitted. By contrast, the i.i.d. estimate computes a comparability probability for two independent generic points under a product model with continuous marginals. It is therefore best interpreted as a stylized baseline for unstructured random sets, not as a model of actual curated archives produced by optimization algorithms.

This distinction matters because curated archives violate the assumptions of the proxy model in systematic ways. They are filtered by nondominance, shaped by the search dynamics, and often correlated across both points and coordinates. The i.i.d. model is therefore not a prediction for archive behavior; it is a reference calculation that quantifies how incomparability scales with dimension in an idealized random setting. The deterministic archive theorem, by contrast, is exact and requires no distributional assumptions. The empirical section was included precisely to keep these roles separate: it measures comparison-count reductions in controlled streams and reports the two diagnostics that explain those reductions, rather than treating the i.i.d. formula as a direct model of archive dynamics.

**Practical integration and witness-seeking traversal.** The modular nature of the rule suggests combining it with archive organizations or traversal heuristics that tend to expose likely witnesses early. Since a witness for an incoming candidate $q$ is a point $w$ satisfying $q \prec w$, it is natural to visit first those archive elements that already appear large relative to $q$ in several coordinates. For example, if the archive is presorted or indexed by selected objective coordinates, one may prioritize regions whose stored values are already high compared with those of $q$, since such regions are more plausible locations for witnesses.

More generally, one may use inexpensive surrogate keys, that is, cheap proxy scores used only to rank archive elements by witness-likelihood before performing full Pareto comparisons. Examples include the number of coordinates in which $q_i \leq u_i$ already holds, the minimum coordinatewise margin $\min_i(u_i - q_i)$, or coarse

surplus scores such as $\sum_i (u_i - q_i)$. None of these quantities certifies $q \prec u$; their role is only to bring more plausible witnesses earlier in the scan.

Likewise, one may maintain coarse objective-space partitions, such as grid cells, bins, or tree-based regions, and query first those parts of the archive whose stored summaries suggest that they may contain points coordinatewise larger than $q$. Again, these partitions are not substitutes for exact dominance checks; they are only devices for ordering the search so that witness events, when they exist, are more likely to be encountered early.

These are implementation heuristics rather than consequences of the theory. The propagation property guarantees that, once a witness has been found, later comparison checks of the appropriate form may be skipped without changing the returned updated archive. It does not, however, identify an optimal policy for locating that witness.

It is equally important not to overstate what follows from the present analysis. The results justify only the following claim: if a witness is encountered earlier in the scan, then the pruning rule is activated earlier, and a larger suffix of comparison checks of the relevant form becomes eligible for elimination. They do *not* show that the first witness encountered is always the witness whose discovery leads to the largest eventual reduction in total comparisons, nor do they identify an optimal policy for witness discovery. The design and analysis of scan orders or archive organizations that systematically maximize pruning or deletion effects therefore remain separate questions for future work. The witness-rich ordering used in Section 4.1 should be understood in exactly this specific sense: it is a synthetic device for illustrating that witness-oriented traversal can materially affect practical savings, not a claim of optimal traversal design.

**Processing cost versus objective-generation cost.** The present results reduce only the cost of processing objective vectors under the Pareto order; they do not reduce the cost of generating those vectors. If objective evaluation is substantially more expensive than archive maintenance—for example, because each candidate requires a costly simulation, a high-fidelity solve, or a large-scale training or validation step—then even a substantial reduction in dominance-comparison counts may translate into only a modest end-to-end wall-clock improvement. By contrast, when objective vectors are relatively inexpensive to obtain, when candidates are produced at high throughput, or when the maintained archive or candidate pool is large enough that repeated insertion or sorting becomes a nontrivial share of runtime, reducing the number of dominance tests can have a more visible computational effect.

This tradeoff in relative cost may also shift over time. Objective generation may become cheaper not only through greater parallelization or amortization, but also through improved modeling practice: better problem formulations, stronger surrogate models, caching and reuse of evaluations Xu et al. (2016), batching strategies Wu & Frazier (2016); Huang et al. (2019), warm starts (that is, initializing a new solve from a previous nearby solution or solver state) CVXPY Authors (2026), and more efficient learning pipelines can all reduce the marginal cost of obtaining objective vectors. As that happens, order-processing may account for a larger fraction of practical runtime, and comparison-reduction mechanisms of the kind studied here may correspondingly become more relevant. The most accurate description of the present contribution is therefore as a reduction in the dominance-processing workload of archive maintenance, not as a blanket claim about total optimization time across all application domains.

**Archive growth and worst-case behavior.** The post-witness pruning rule reduces comparison workload during archive maintenance, but it does not prevent archive growth. In worst-case regimes, many processed points may remain mutually nondominated, and the archive may become large. In the linear-scan insertion model studied here, the cumulative worst-case cost of repeated insertion therefore remains quadratic in the number of processed points. The rule should therefore be interpreted as a correctness-preserving reduction in dominance-testing workload, not as a complete remedy for archive-growth bottlenecks. When archive size itself becomes the dominant issue, the rule is best viewed as one component within a broader implementation strategy that may also include specialized archive data structures for dynamic Pareto updates (such as Quad-trees or ND-trees), bounded or truncating archive policies (such as crowding-distance- or truncation-based selection), and structured archiving schemes based on adaptive grids or decomposition of the objective space

into subregions Mostaghim & Teich (2005); Jaszkiewicz & Lust (2018); Knowles & Corne (2004); Zitzler et al. (2001); Deb et al. (2002).

**Beyond exact strict Pareto comparisons.** The same order-theoretic reasoning based on transitivity and incomparability extends beyond exact strict Pareto comparisons whenever the operative pruning relation is transitive and incomparability is defined with respect to that same relation. This includes the weak- and $\varepsilon$-dominance settings discussed earlier, provided the chosen comparison rule satisfies those properties.

For noisy objective evaluations, however, the relevant issue is no longer only comparison reduction, but the reliability of the comparisons themselves. The paper treats two distinct extensions. In the certified-comparison framework, pruning is allowed only when the available uncertainty bounds certify the required relation; on the relevant coverage event, the resulting pruning and deletion decisions are then consistent with the underlying true objective vectors. In the repeated-confirmation framework, by contrast, the guarantee is probabilistic rather than exact: under the stated error-rate and independence assumptions, repeated confirmations control the probability of an incorrect skip at a single pruning event. These two extensions should therefore be interpreted differently. The certified framework yields conditional exactness on the stated coverage event, whereas the repeated-confirmation framework yields local probabilistic error control.

**Relation to full nondominated-sorting pipelines.** The present paper studies archive insertion because it is the cleanest setting in which the post-witness pruning rule can be isolated, formalized, and analyzed. In full nondominated-sorting pipelines, the same logic is most naturally relevant inside insertion-like or archive-maintenance subroutines rather than as a stand-alone replacement for an entire sorter. The paper therefore does not claim a new complete nondominated-sorting algorithm. Rather, it identifies a reusable pruning primitive that can be embedded into comparison-driven subroutines wherever the required dominance tests arise. A fuller integration into modern nondominated-sorting implementations is a natural next step, especially if accompanied by systematic benchmarks reporting both dominance-predicate counts and wall-clock time across a range of archive sizes, objective dimensions, correlations, and data-generation regimes.

## 6    Conclusion

The central point of this paper is that strict Pareto incomparability is not merely a description of a trade-off between two objective vectors; it has exact logical consequences. If two vectors are fixed as strictly incomparable, then knowledge of one strict dominance relation involving one of them constrains what can still hold relative to the other. In particular, if $u \parallel v$, then there exists no third vector $q$ such that $u \prec q \prec v$ or $v \prec q \prec u$. In the archive-insertion setting, this impossibility of a two-step strict-dominance bridge between strictly incomparable vectors yields a precise pruning rule: once a witness $w \in S$ with $q \prec w$ has been found in a mutually nondominated archive $S$, the remaining tests of the form "$u \prec q$?" cannot succeed. This pruning rule is therefore not heuristic; it removes only comparisons whose failure is already forced by the order structure together with the archive invariant. Its computational consequence is a reduction in dominance-processing workload whenever such witness events occur.

Viewed in this way, the contribution of the paper is to expose a form of *order-induced redundancy*. The same local information that certifies one strict dominance relation also certifies the impossibility of a family of others. The paper formalizes this redundancy as the *propagation property*, turns it into a correctness-preserving pruning rule for archive insertion, and then studies its computational effect through both an i.i.d. proxy calculation and an empirical illustration. Geometrically, the same propagation property can be read through strict down-sets: if $q \prec v$, then $\downarrow q \subset \downarrow v$, and $u \prec q$ would force $u \in \downarrow v$, equivalently $u \prec v$, contradicting $u \parallel v$. What emerges is a consistent picture: a structural consequence of strict Pareto dominance becomes operational once it is made explicit, and, in the settings treated in the paper, it yields an exact reduction in dominance-processing workload whenever witness events occur.

The scope of the result is correspondingly specific. The pruning rule acts on dominance-processing workload; it does not reduce objective-evaluation cost, does not prevent archive growth, and does not by itself replace a complete nondominated-sorting framework. Its strongest deterministic form applies when a mutual-nondominance invariant is maintained. In the certified noisy-comparison setting, the same pruning logic

remains conditionally exact on the relevant coverage event for the certified comparisons used by the update. By contrast, the repeated-confirmation variant provides only local probabilistic error control under the stated noise assumptions. Within that scope, the paper identifies a class of dominance tests that cannot affect correctness in the deterministic or certified settings and may therefore be omitted.

# 7 Future work

Several future research directions follow naturally. One is systematic integration into comparison-driven archive-maintenance and nondominated-sorting implementations, together with benchmarks that separate predicate-count savings from end-to-end runtime effects. A second is the design and analysis of scan-order or archive-organization policies that expose valid witnesses earlier and thereby increase realized pruning, without changing the logical guarantees of the pruning rule. A third is a fuller treatment of stochastic comparison settings beyond the certified regime considered here. Relevant cases include *correlated noise*, where repeated comparison outcomes are dependent and the simple independent-confirmation error bound no longer applies directly; *adaptive sampling*, where the information budget assigned to a candidate depends on intermediate outcomes and may therefore interact with witness discovery; and *uncertainty-aware archive updates*, where insertion, deletion, or abstention decisions are based on confidence bounds, posterior uncertainty, or related reliability measures rather than exact objective vectors alone. Extending the pruning rule to such settings requires more than repeated confirmation: it requires a framework for tracking how uncertainty, dependence, and sequential sampling decisions affect the logical safety of witness-based pruning.

More broadly, the paper suggests that other simple but structurally exact consequences of partial orders on $\mathbb{R}^n$ may still be hiding in plain sight, waiting to be turned from logical facts into useful computational primitives.

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
