# OpenReview forum: "A Propagation Property of the Pareto Order with Applications to Multiobjective Optimization"
_TMLR — Rejected by TMLR_

### Review · Reviewer_eE2u · 2025-11-18

**Summary Of Contributions:**

The core idea of this paper is that, for a set of non-dominated points, if we find a new point that dominates one of them, then the other points in the set cannot dominate this new point. Either the new point also dominates the remaining points, or it is non-dominated with respect to them. Next, this idea can be used to accelerate the applications, such as the standard incremental Pareto archive insertion routine.

The proof is mathematically correct, but it is actually just an elementary consequence of the transitivity of the Pareto order. The paper spends significant space on "geometric proofs" and "strict down-sets", and gives an obvious conclusion. I think it is quite trivial. Also, there is no empirical evidence.

**Audience:**

No

**Audience Explanation:**

The conclusion is quite trivial; it is simply a direct consequence of the transitivity of Pareto order/dominance.

In addition, although this conclusion can be applied to routines such as the standard incremental Pareto archive insertion procedure, I question whether people actually use the standard version in practice. There should be many more efficient variants available. Why does the paper not mention or compare against them?

**Broader Impact Concerns:**

There are no impact concerns.

**Claims And Evidence:**

Yes

**Claims Explanation:**

The proof is just a straightforward application of the transitivity of Pareto order, which is simple and obvious.

**Requested Changes:**

The contribution is too trivial for publication, and I would recommend rejection. The conclusion is quite trivial; it is simply a direct consequence of the transitivity of Pareto order/dominance.

---

> ### Author Response · Authors · 2026-02-28
> **Response to Reviewer eE2u**
>
> We thank the reviewer for taking the time to read the manuscript and for confirming that the mathematical statements and proofs are correct. We address the concerns regarding contribution, novelty, exposition choices, empirical illustration, and practical relevance below.
>
> ### 1. On the characterization of the core contribution
>
> >“The core idea of this paper is that, for a set of non-dominated points, if we find a new point that dominates one of them, then the other points in the set cannot dominate this new point…”
>
> This is a reasonable high-level description of the archive-level implication discussed in the manuscript (the Discussion/Conclusion), namely the post-witness elimination effect: let $S\subset\mathbb{R}^k$ be a mutually nondominated archive and let $q\in\mathbb{R}^k$ be a new candidate to be inserted. Once a \emph{witness} $w\in S$ with $q\prec w$ is found (i.e., $q$ is dominated by an archive point), all subsequent dominance checks of the form ``$u\prec q$?'' for $u\in S\setminus\{w\}$ are provably unnecessary.
>
> The manuscript does not merely suggest that the propagation property can be used to accelerate dominance-testing in archive maintenance. It makes the propagation property explicit and states it as a standalone result, then uses it to derive a post-witness elimination rule, formalize a concrete archive-insertion routine (pseudocode), and prove its correctness: the routine keeps the archive unchanged exactly when the candidate is dominated; otherwise, it inserts the candidate and removes exactly the points it dominates. In addition, the i.i.d.\ analysis gives a closed-form estimate of the expected post-witness fraction of dominance checks that can be skipped under a standard i.i.d.\ continuous proxy model.
>
> ---
>
> ### 2. On the concern that the result is “just an elementary consequence of transitivity”
>
> >“The proof is mathematically correct, but it is actually just an elementary consequence of the transitivity of the Pareto order… I think it is quite trivial.”
>
> We agree that, once explicitly formulated, the propagation property follows logically from transitivity. The manuscript makes this dependence transparent and includes a concise transitivity-based proof for clarity. The manuscript does not claim the rediscovery of transitivity, but the identification, isolation, and formalization of a specific structural consequence—namely, the impossibility of a two-step dominance bridge between incomparable points—and its systematic exploitation as a pruning rule in archive maintenance and nondominated sorting. Triviality would require showing that this specific consequence has already been explicitly stated and used as a pruning primitive in Pareto archive maintenance / nondominated sorting; we are not aware of such prior explicit formulation and use.
>
> ---
>
> ### 3. On the absence of empirical evidence, and use of multiple proofs
>
> >“The paper spends significant space on ‘geometric proofs’ and ‘strict down-sets’, and gives an obvious conclusion. I think it is quite trivial. Also, there is no empirical evidence.”
>
> - The manuscript presents multiple proofs intentionally: each proof highlights a different viewpoint (transitivity/order-theoretic, coordinatewise/algebraic, and down-set/geometric), for clarity and making the order-theoretic mechanism explicit.
> - The correctness of the elimination rule follows from structural properties of the dominance relation and is established formally. Any empirical component would therefore serve to illustrate the magnitude of comparison savings in practice, rather than to validate the underlying correctness. However, we agree that including a computational illustration demonstrating the reduction in dominance comparisons under controlled inputs would strengthen the presentation.
>
> ---
>
> ### 4. On the question of whether standard routines are used in practice
>
> >“I question whether people actually use the standard version in practice. There should be many more efficient variants available.”
>
> We agree that there are alternative archive-maintenance and nondominated-sorting algorithms beyond the simplest incremental routine, and the manuscript does not claim otherwise. However, the proposed elimination rule operates at the level of dominance testing itself. Regardless of the specific archive structure or sorting strategy employed, methods that maintain mutually nondominated sets fundamentally rely on dominance predicates. The propagation property reduces a subset of these dominance checks once a witness is identified, without altering correctness or asymptotic guarantees. In this sense, the contribution is orthogonal to the choice of archive-maintenance framework and can be integrated into existing variants rather than being tied to a particular “standard” implementation.
>
>
> We thank the reviewer again for their time and consideration.

---

### Review · Reviewer_KL5H · 2026-01-08

**Summary Of Contributions:**

This paper investigates a theoretical property of the Pareto dominance order in multiobjective optimization and leverages it to improve algorithms for nondominated sorting and Pareto archive maintenance. The main result (Theorem 1) establishes a propagation property of strict dominance: if two solutions u and v are mutually nondominated (incomparable), then no third point q can exist such that u ≺ q ≺ v or v ≺ q ≺ u. In other words, there is no “two-step” dominance chain bridging two incomparable points. Using this property, the authors derive a practical post-witness elimination rule for Pareto archives: once a witness point w is found in the archive such that q ≺ w (the new candidate q is dominated by w), then no other point u in the archive can dominate q. This means all further “u ≺ q?” checks can be skipped after the first witness, requiring only the checks of “q ≺ u?” for removing points dominated by q. The paper presents this idea with multiple supporting components: the authors give three proofs of the structural theorem (an algebraic derivation, a one-line proof via transitivity, and a geometric argument using down-sets) to build intuition. They provide pseudocode (Algorithm 1) for an improved archive insertion procedure that implements the rule, and prove its correctness (Theorem 2) for maintaining a mutually nondominated set. Furthermore, a probabilistic analysis under an i.i.d. random input model yields a closed-form estimate of the expected savings in dominance comparisons. Notably, the analysis predicts that the fraction of dominance checks pruned after finding the first witness is about 1 – 2^1–k (for k objectives). This indicates significant efficiency gains, especially as the number of objectives k grows (e.g. with large k, the probability of any second point dominating q after a witness is found becomes very small). The manuscript also discusses how the propagation property and elimination rule extend to variants of dominance (weak dominance, ε-dominance, and even noisy or approximate comparisons) with appropriate adjustments, underscoring the generality of the approach.

**Audience:**

Yes

**Audience Explanation:**

Multiobjective Optimization is an important field in Machine Learning.

**Broader Impact Concerns:**

No broader impact concerns.

**Claims And Evidence:**

Yes

**Claims Explanation:**

It provides the proof in the paper.

**Requested Changes:**

Despite its strengths, the paper has a few weaknesses or areas where it could be improved. The most notable limitation is the lack of empirical evaluation. All results presented are theoretical or analytical; the paper does not include experiments or computational benchmarks to demonstrate the actual performance gains of the proposed method. As a result, the practical impact, while strongly argued, remains somewhat speculative until validated. The authors do acknowledge this and suggest that future work will include empirical studies and integration into sorting routines. Still, the review process could benefit from at least a small-scale experiment or an illustrative example (for instance, inserting a series of points into an archive with and without the elimination rule and comparing the number of comparisons made). Such an example would concretely demonstrate the benefit and help readers gauge the magnitude of improvement in realistic scenarios. Another minor weakness is that the core propagation property might appear too obvious to some readers. Since it essentially follows from the transitivity of the dominance relation (as the authors themselves note via a one-line proof), it might raise the question of whether this is a new discovery or a rediscovery of a known fact. The paper would benefit from explicitly addressing this perspective — perhaps by emphasizing that although the property is logically inherent to any partial order, recognizing and using it in Pareto optimization is non-trivial and previously overlooked. In other words, the novelty is not in the mathematics of the property but in its application and consequences. Making this distinction clear can manage reader expectations and preempt the impression that the authors are claiming something obvious as new.

In terms of practical integration, the paper could offer more guidance. The discussion section provides some insight (for example, suggesting that one could cluster points in data structures to quickly find incomparable regions once a witness is found, or that “witness-first” scanning orders are heuristicly justified), but these ideas are not fleshed out. Offering a bit more detail on how to choose an order to scan the archive for a witness (e.g. sorting archive points by some dominance heuristic or partial order of objectives) or referencing any preliminary tests of such strategies would strengthen the paper’s practical relevance. Additionally, a complexity analysis of the proposed algorithm in best, average, and worst cases could be added for completeness. It’s implied that worst-case remains O(N^2) and best-case approaches O(N) in the presence of an early witness, but quantifying this or discussing scenarios that achieve near-best-case would be instructive. These suggestions are relatively minor and intended to enhance the paper. They do not detract from the correctness or relevance of the work, but addressing them could improve the manuscript’s robustness and impact. In particular, an empirical component (even if simple) and a clearer connection from theoretical insights to implementation guidelines would help fully convince the audience of the value of the propagation property in day-to-day multiobjective optimization tasks. Overall, I encourage the authors to incorporate these improvements, as they would make an already strong paper even more compelling.

---

> ### Author Response · Authors · 2026-02-24
> **Response to Reviewer KL5H**
>
> We thank the reviewer for the careful reading and constructive feedback. We appreciate the clear articulation of both the strengths and areas for improvement. We address the main concerns below.
>
>
> ### 1. On the absence of empirical evaluation
>
> >“The most notable limitation is the lack of empirical evaluation… As a result, the practical impact, while strongly argued, remains somewhat speculative until validated… the review process could benefit from at least a small-scale experiment.”
>
> We thank the reviewer for this suggestion. We agree that while the manuscript establishes correctness and comparison-count savings formally, the observed wall-clock impact in concrete implementations can depend on regime- and implementation-specific factors (e.g., archive sizes encountered in practice, data-structure overheads, and hardware/parallelization effects). In that sense, an empirical illustration would strengthen the presentation by demonstrating the practical magnitude under representative settings.
>
> In a revision, we are willing to add a small-scale computational illustration as the reviewer suggests (e.g., inserting sequences of points into an archive with and without the elimination rule), to make the practical effect more directly observable. This would complement the theoretical analysis without changing the scope or claims of the manuscript.
>
> ---
>
> ### 2. On the concern that the property may appear “too obvious”
>
> >“Since it essentially follows from the transitivity of the dominance relation … it might raise the question of whether this is a new discovery or a rediscovery of a known fact… The novelty is not in the mathematics of the property but in its application and consequences.”
>
> We agree with this framing and thank the reviewer for stating it clearly.
>
> Indeed, once the propagation property is explicitly formulated, it follows directly from the transitivity of the Pareto (dominance) order; the manuscript includes a concise transitivity-based proof to make this logical dependence explicit. The contribution is therefore not the rediscovery of transitivity, but the identification and formalization of a specific structural consequence—namely, the impossibility of a two-step dominance chain bridging two incomparable points—and the demonstration that this consequence yields a correctness-preserving post-witness elimination rule for archive maintenance (and related dominance-testing routines).
>
> To the best of our knowledge, this particular consequence has not been explicitly isolated and then exploited in this form as an elimination mechanism in the Pareto archive maintenance / nondominated sorting literature. Following the reviewer’s suggestion, we will revise the manuscript to make this positioning more explicit, emphasizing that the novelty lies in isolating the structural consequence and deriving/using its algorithmic implications, rather than in the underlying abstract order-theoretic axiom.
>
> ---
>
> ### 3. On practical integration and scanning order
>
> >“The paper could offer more guidance … how to choose an order to scan the archive for a witness …”
>
> We thank the reviewer for this helpful suggestion. The manuscript’s main goal is to establish the structural elimination rule and its correctness guarantees; consequently, it intentionally does not prescribe a particular witness-search policy, since the scan order is an implementation choice that is orthogonal to correctness. However, we agree that readers would benefit from clearer practical guidance on how the rule can be integrated into existing archive-maintenance and sorting pipelines.
>
> In revision, we will add a short discussion clarifying how the rule plugs into standard archive-maintenance/sorting pipelines and how generic “witness-first” scan orders can improve early witness discovery, without changing assumptions or expanding the algorithmic scope.
>
> ---
>
> ### 4. On complexity analysis
>
> >“A complexity analysis of the proposed algorithm in best, average, and worst cases could be added…”
>
> We agree that a clearer articulation of complexity regimes would strengthen the manuscript.
>
> The worst-case asymptotic cost is unchanged: per insertion of a candidate into an archive $S$ (with $|S|$ points in $\mathbb{R}^k$), the procedure remains $\mathcal{O}(|S|)$ dominance tests, i.e., $\mathcal{O}(|S|k)$ coordinate comparisons, and thus does not alter worst-case bounds. However, once a witness $w \in S$ with $w \prec q$ is found early, all subsequent checks of the form ``$u \prec q$?'' are provably unnecessary, improving best-case behavior and reducing constants in practice.
> Moreover, our i.i.d.\ analysis already quantifies the expected fraction of pruned dominance checks as a function of the objective dimension $k$.
>
> In revision, we will make these worst-/best-/expected-case statements explicit in one place.
>
> We thank the reviewer again for the constructive suggestions, which we believe will further strengthen the manuscript.

---

### Review · Reviewer_dcpG · 2026-02-02

**Summary Of Contributions:**

This paper proposes a novel rule for archive maintenance in multi-objective optimization.
It is based on the observation that for a partial order $\prec$, whenever $u \not \prec v$ and $v \not \prec u$, there is no $q$ such that $q \prec v$ and $q \prec u$.
In multi-objective optimization, one typically aims to find the Pareto set (i.e., a set of points where one cannot improve all function values simultaneously).
In multi-objective optimization, one aims to find this set by adding novel candidates to it/removing points from it.
As this set contains points that satisfy neither $u \prec v$ nor $v \prec u$, this implies that for any point $q$ (called a witness), one can remove all elements $a$ in the Pareto set such that $q \prec a$, effectively pruning the Pareto set using this witness.
In a case study on randomly generated points, it is shown that this rule can be used to reduce by a large factor the number of required comparisons in computing the Pareto set.

**Audience:**

Yes

**Audience Explanation:**

This paper proposes a different point of view on multi-objective optimization, based on trying to limit the size of the Pareto set estimates using the post-witness elimination rule. This should be of interest to the TMLR audience, as it allows for reducing the number of comparisons performed in maintaining an archive in multi-objective optimization by a large factor.

However, in multi-objective optimization, I believe that the most costly step lies in evaluating the function's values and finding the points to which these functions should be evaluated, not in maintaining an archive. This distinction is not mentioned explicitly in the paper, and there is no discussion on how much time this may gain when combined with actual multi-objective optimization methods.

In this regard, it seems that the contribution may be of interest either (i) in multi-objective problems where evaluating the functions' values is not expensive, or (ii) where the number of candidates for the Pareto set easily becomes very large. The existence and relevance of these problems are not clear to the reviewer, which should be made more explicit to guarantee the interest of TMLR audience in these results. Nonetheless, it is likely that any multi-objective optimization method can benefit from this, although the gains may be imperceptible or marginal in most practical cases.

**Broader Impact Concerns:**

This work does not raise any particular ethical questions.

**Claims And Evidence:**

Yes

**Claims Explanation:**

All the results from the paper are provided with proofs (even three proofs for the main result):
- The proof that if two points are not mutually dominated, there is no "in-between point" is correct.
- The application for updating the Pareto set estimate in multi-objective is correct.
- The application to the iid sampling case is also correct.

Extensions to weak dominance and $\epsilon$ dominance are discussed. Applications to problems where the points are computed with some noise are mentioned, although without proof or many details.

**Requested Changes:**

There are a few things that could be done to ensure the relevance of the paper to the TMLR community:
- Extending the discussion on "noisy evaluation/comparison", which is only mentioned as a remark at the end of Section 4, would really help. In particular, what would happen if the objective functions are empirical risk, and are only evaluated on a fraction of the dataset?
- Further extending on the noisy comparison case, could there be a way to correct previous errors in the case where many elements were discarded due to noisy evaluation? Is there a chance that all current candidates could be eliminated due to a very noisy function evaluation, which would be extremely optimistic in all of the objective's problems simultaneously?
- The authors should discuss the settings in which the gains may be significant: in particular, when evaluating the function is the most costly, then maintaining the archive seems to have negligible cost in comparison with evaluating the value of the points themselves. Are there some cases where the archive may grow in an uncontrolled way such that even if the comparisons are inexpensive, it becomes a bottleneck in the optimization process?
- There is no discussion in the paper on how one could find a witness such that the archive can be reduced a lot: do the proposed results allow for finding new rules to propose witnesses that may help in eliminating many points from the archive?

---

> ### Author Response · Authors · 2026-02-24
> **Response to Reviewer dcpG (Part 1 of 2)**
>
> We thank the reviewer for the thoughtful and detailed comments. We address each concern below.
>
> ### 1. On the relative cost of objective evaluation versus archive maintenance
>
> > “In multi-objective optimization, I believe that the most costly step lies in evaluating the function's values and finding the points to which these functions should be evaluated, not in maintaining an archive. This distinction is not mentioned explicitly in the paper.”
>
> We agree that in many large-scale ML settings, objective evaluation can dominate wall-clock runtime, and the manuscript does not claim to reduce that cost. Our contribution instead targets a distinct and complementary component of the multiobjective optimization pipeline: the structural cost of dominance testing and archive maintenance once objective vectors are available.
> Even in regimes where evaluation dominates, archive maintenance remains operationally relevant for two reasons. First, it is invoked after essentially every evaluation—each new candidate must be compared to the archive—so its cost accumulates over many iterations and can become non-negligible when archives are large, the number of objectives is moderate-to-large, or candidate throughput is high. Second, the relative weight of this structural cost is not fixed. Improvements in evaluation efficiency (e.g., batching, surrogate modeling, parallelization, or hardware acceleration) reduce the evaluation component and may shift bottlenecks toward downstream structural operations. A reduction in dominance-comparison workload is therefore a model-agnostic structural improvement whose relevance does not depend on a particular evaluation regime.
>
> To avoid any ambiguity, we will revise the manuscript to explicitly separate (i) evaluation cost from (ii) archive/dominance cost, and to state that our guarantees concern dominance-comparison reduction with correctness preserved; the extent of end-to-end runtime improvement depends on the regime and implementation
>
> ---
>
> ### 2. On the absence of explicit runtime quantification within full optimization pipelines
>
> > “…there is no discussion on how much time this may gain when combined with actual multi-objective optimization methods.”
>
> The manuscript provides a formal reduction in dominance comparisons and an analytical estimate of expected pruning under an i.i.d. model. It does not translate this reduction into wall-clock runtime improvements within a specific optimization algorithm.
> This was intentional. Total runtime depends on several interacting components, including objective evaluation, candidate generation, and archive update. The relative contribution of dominance testing varies by regime. Rather than introduce application-specific timing assumptions, the manuscript isolates and analyzes the structural comparison component directly.
>
> We agree that this scope should be stated more clearly. In revision, we can emphasize that the guarantee concerns comparison-count reduction, and that overall runtime impact is regime-dependent.
>
> ---
> ### 3. On the relevance of regimes where dominance comparison cost is significant
>
> >“…the contribution may be of interest either (i) in problems where evaluating the functions' values is not expensive, or (ii) where the number of candidates for the Pareto set easily becomes very large. The existence and relevance of these problems are not clear…”
>
> We appreciate this request for clearer contextual positioning.
>
> The key point is that the proposed pruning rule targets a structural cost that grows with (a) how often the archive is updated and (b) how large the archive (or candidate pool) becomes—independent of why candidates are generated.
> Such regimes are common and well-studied in the multiobjective optimization literature. Examples include: (i) many-objective optimization (moderate-to-large k), where dominance testing is repeatedly performed but comparability is rare, so naive procedures spend substantial effort on unsuccessful dominance checks; (ii) population-based methods and external archives that maintain (or repeatedly filter) large nondominated sets over many generations; and (iii) incremental/streaming Pareto updates where nondominated filtering is performed continuously as new points arrive. In these settings, dominance-comparison counts—and their reduction—are explicitly treated as an algorithmic optimization target in the nondominated sorting and archive-maintenance literature.
>
> We agree that the manuscript should more clearly describe these regimes. We will revise the discussion to specify where structural dominance reduction is most likely to have measurable impact.
>
> (continued in next comment)

---

> > ### Author Response · Authors · 2026-02-24
> > **Response to Reviewer dcpG (Part 2 of 2)**
> >
> > (continued from previous comment)
> >
> > ### 4. On noisy evaluation and empirical risk settings
> >
> > >“Extending the discussion on ‘noisy evaluation/comparison’… what would happen if the objective functions are empirical risk…?”
> >
> > The propagation property is deterministic and assumes a consistent dominance predicate. If objective values are noisy estimates (e.g., empirical risk computed on subsets of data), dominance decisions may be misclassified.
> >
> > The elimination rule does not introduce additional structural risk beyond what already exists in dominance-based archive procedures. Any erroneous dominance decision due to noise would affect standard archive insertion in the same way.
> >
> > We agree that this distinction should be stated more explicitly. In revision, we will:
> >
> > -	Clarify the assumption of consistent dominance predicates;
> > -	Discuss how stochastic evaluation impacts dominance reliability;
> > -	Emphasize that robustness to noise is a property of the dominance predicate, not of the elimination rule itself.
> >
> > ---
> >
> > ### 5. On the possibility of over-pruning under noisy dominance
> >
> > >“Is there a chance that all current candidates could be eliminated due to very noisy evaluation?”
> >
> > Under correct strict dominance, the elimination rule preserves the mutually nondominated property of the archive. Under noisy dominance misclassification, erroneous pruning could occur in any dominance-based archive mechanism.
> > The proposed rule does not introduce new failure modes; it relies on the same dominance decisions as standard procedures. We will clarify this explicitly in the revised manuscript.
> >
> > ---
> >
> > ### 6. On witness discovery and scanning strategies
> >
> > >“There is no discussion… on how one could find a witness such that the archive can be reduced a lot…”
> >
> > The manuscript establishes a correctness-preserving elimination rule conditional on discovering a witness. It does not prescribe a specific scanning strategy for identifying such a witness.
> >
> > We agree that clarifying this separation would improve the presentation. In revision, we will state explicitly that the propagation result guarantees pruning once a witness is identified, while witness-selection heuristics are orthogonal design choices outside the scope of the core structural contribution.
> >
> > ---
> >
> > ### Summary of Revisions
> >
> > In response to the reviewer’s suggestions, we will:
> >
> > -	Explicitly distinguish evaluation cost from structural dominance cost;
> > -	Clarify that runtime gains are regime-dependent;
> > -	Expand discussion of regimes where comparison reduction is significant;
> > -	Strengthen discussion of noisy dominance assumptions and limitations;
> > -	Clarify the role of witness discovery strategies.
> >
> > We again thank the reviewer for these constructive suggestions, which will improve the clarity and positioning of the manuscript.

---

### Decision · Action_Editor_ePtC · 2026-04-16

**Recommendation:** Reject

**Audience:**

No

**Audience Explanation:**

While multi-objective optimization (MOO) itself is a significant subfield of machine learning, the reviewers and the AE agree that the specific contributions in this work may lack sufficient interest for the TMLR community.

Reviewers still have concerns around if the main theoretical results are a restatement of the transitivity of partial orders. Moreover, there is a disconnection between the theoretical saving and practical benefits in ML. For instance, in MOO, the evaluation and optimization of the cost objective function dominates the algorithms compared with Pareto point maintenance. I would encourage the authors to further establish stronger bridges to implications of practical ML applications or existing MOO variants.

**Claims And Evidence:**

Yes

**Claims Explanation:**

The theoretical arguments (the "propagation property" of Pareto dominance) in the submission are supported by proofs and mathematically sound, as agreed by all the reviewers. The authors also provide probabilistic analysis and other extensions. However, the language and the organization of the paper in its current form may make it difficult for readers from the ML community to access the claims.

**Resubmission Of Major Revision:**

The authors may consider submitting a major revision at a later time.